# Behavioral evidence for memory replay of video episodes in the macaque

**Shuzhen Zuo[1†], Lei Wang[1†], Jung Han Shin[2], Yudian Cai[1], Boqiang Zhang[1], Sang Wan Lee[3], Kofi Appiah[4], Yong-di Zhou[5], Sze Chai Kwok[1,6,7]\***

[1]Shanghai Key Laboratory of Brain Functional Genomics, Key Laboratory of Brain Functional Genomics Ministry of Education, School of Psychology and Cognitive Science, East China Normal University, Shanghai, China; [2]Program of Brain and Cognitive Engineering, Korea Advanced Institute of Science and Technology, Daejeon, Republic of Korea; [3]Department of Bio and Brain Engineering, Korea Advanced Institute of Science and Technology, Daejeon, Republic of Korea; [4]Department of Computer Science, University of York, York, United Kingdom; [5]School of Psychology, Shenzhen University, Shenzhen, China; [6]Shanghai Key Laboratory of Magnetic Resonance, East China Normal University, Shanghai, China; [7]NYU-ECNU Institute of Brain and Cognitive Science at NYU Shanghai, Shanghai, China

**\*For correspondence:**
sze-chai.kwok@st-hughs.oxon.org

[†]These authors contributed equally to this work

**Competing interests:** The authors declare that no competing interests exist.

**Abstract** Humans recall the past by replaying fragments of events temporally. Here, we demonstrate a similar effect in macaques. We trained six rhesus monkeys with a temporal-order judgement (TOJ) task and collected 5000 TOJ trials. In each trial, the monkeys watched a naturalistic video of about 10 s comprising two across-context clips, and after a 2 s delay, performed TOJ between two frames from the video. The data are suggestive of a non-linear, time-compressed forward memory replay mechanism in the macaque. In contrast with humans, such compression of replay is, however, not sophisticated enough to allow these monkeys to skip over irrelevant information by compressing the encoded video globally. We also reveal that the monkeys detect event contextual boundaries, and that such detection facilitates recall by increasing the rate of information accumulation. Demonstration of a time-compressed, forward replay-like pattern in the macaque provides insights into the evolution of episodic memory in our lineage.

## Introduction

Accumulating evidence indicates that non-human primates possess the ability to remember temporal relationships among events (*Templer and Hampton, 2013*; *Gower, 1992*; *Charles et al., 2004*). The apes can remember movies based on the temporal order of scenes (*Morimura and Matsuzawa, 2001*) and keep track of the past time of episodes (*Martin-Ordas et al., 2010*), whereas macaque monkeys possess serial (*Terrace et al., 2003*) and ordinal positions expertise for multi-item lists (*Chen et al., 1997*) and are able to categorize sequences of fractal images by their ordinal number (*Orlov et al., 2000*). However, keeping track of and remembering the positional coding, and forming associative chaining (*Templer et al., 2019*; *Long and Kahana, 2019*) of lists of arbitrary items might diverge from how a semantically linked, temporally relational representation of real-life events is maintained and utilized.

In the human literature, it has been shown that episodes can be replayed sequentially on the basis of learned structures (*Liu et al., 2019*), sensory information (*Michelmann et al., 2019*), and pictorial content (*Wimmer et al., 2019*). These findings suggest the possibility that monkeys can rely on a similar mechanism in recalling events that are linked temporally during temporal order judgement

(TOJ). However, the extent to which mechanisms of temporal order judgement overlap across humans and monkeys remains undefined. One hypothesis is that the macaques can similarly rely on a scanning model for information retrieval – akin to serial replay of episodes – to perform temporal order judgements (*Gower, 1992*; *Charles et al., 2004*). In this model, after encoding streams of events, the animal performs retrieval by replaying the stream of information in a forward direction. In this way, retrieval time (RT) would be positively correlated to the temporal distance between the beginning of the stream and the target location. Recent findings in rodents (*Panoz-Brown et al., 2018*) and in humans of memory replay during cued-recall tasks across fragments of video episodes are characterized, however, by replay that proceeds in a forward manner and is temporally compressed (*Michelmann et al., 2019*). A feature of this more sophisticated mechanism is that memory replay is a fluidic process that allows subjects to skip flexibly across sub-events (*Michelmann et al., 2019*). Subjects can omit non-informative parts of episodes and replay a shorter episode (shorter than physical perception) in memory, which contains less information. This interpretation is supported by other works on the mental simulation of paths (*Bonasia et al., 2016*) and video episodes (*Michelmann et al., 2019*). This latter model constitutes a global compression of parts of episodes (that allows skipping across sub-events) and is regarded as substantially superior to a strict forward-replay mechanism.

In order to simulate the dynamic flow of information that occurs in real-life scenarios, we used naturalistic videos as experimental material to study the mechanism of memory retrieval of event order in the monkeys. These videos are more realistic than the arbitrary items or images that were used in previous studies (*Templer and Hampton, 2013*; *Naya et al., 2017*). We used a temporal order judgement paradigm to examine whether and to what extent the pattern underlying memory retrieval conforms to a time-compressed, forward-replay mechanism. In each trial, monkeys watched a naturalistic video composed of two clips, and following a 2 s retention delay, made a temporal order judgement to choose the frame that was shown earlier in the video between two frames extracted from that video (*Figure 1A*). The two frames were either extracted from the same clip or from two different clips of the video. Given that analyses on response latency can provide insights into the extent to which the monkeys' behavior might conform to the two putative replay models outlined above, we looked into the RT data. By applying representational similarity analyses (RSA), the Linear Approach to Threshold with Ergodic Rate (LATER) model and generalized linear models to the RT data, we examined the presence of replay-like behavioral patterns in the monkeys. Specifically, if monkeys recall the frames by their ordinal positions, this would imply a linear increase in their retrieval times. By contrast, if the memory search entails a complex processing of the content determined by their semantically linked, temporally relational linkage within the cinematic footage, we should observe evidence of some non-linear pattern.

Our results suggested that macaque monkeys might adopt a time-compressed, replay-like pattern to search within the representation of continuous information. This time-compression characteristic refers to durations of memory replay that are significantly shorter than the length of the videos. We found that while both species recall the video content non-linearly, there is an aspect of discrepancy between the two species in which the monkeys do not compress the cinematic events globally as effectively as in humans, whereas human participants possess an ability to skip irrelevant information within the video. Finally, we revealed that the monkeys can make use of context changes to facilitate memory retrieval, thus increasing their rate of information accumulation in a drift diffusion model framework.

## Results

### Human-like forward replay in macaques

All six monkeys learned to perform the temporal order judgement task with dynamic cinematic videos as encoded content (*Figure 1B* left and *Video 1*). The six monkeys performed the task with a significantly above chance level with an overall accuracy of 67.9% ± 1.5% (mean ± SD). The human participants performed the task on average at 92.7% ± 1.2% (*Figure 1—figure supplement 1A*, left). Note that there are two main kinds of TOJ trials: 'within-context' and 'across-context' trials (*Figure 1A*). Here, we are first concerned with the response times (RT) data from 'within-context' trials, which allow us to examine TOJ mechanisms, whereas RT data from 'across-context' trials were

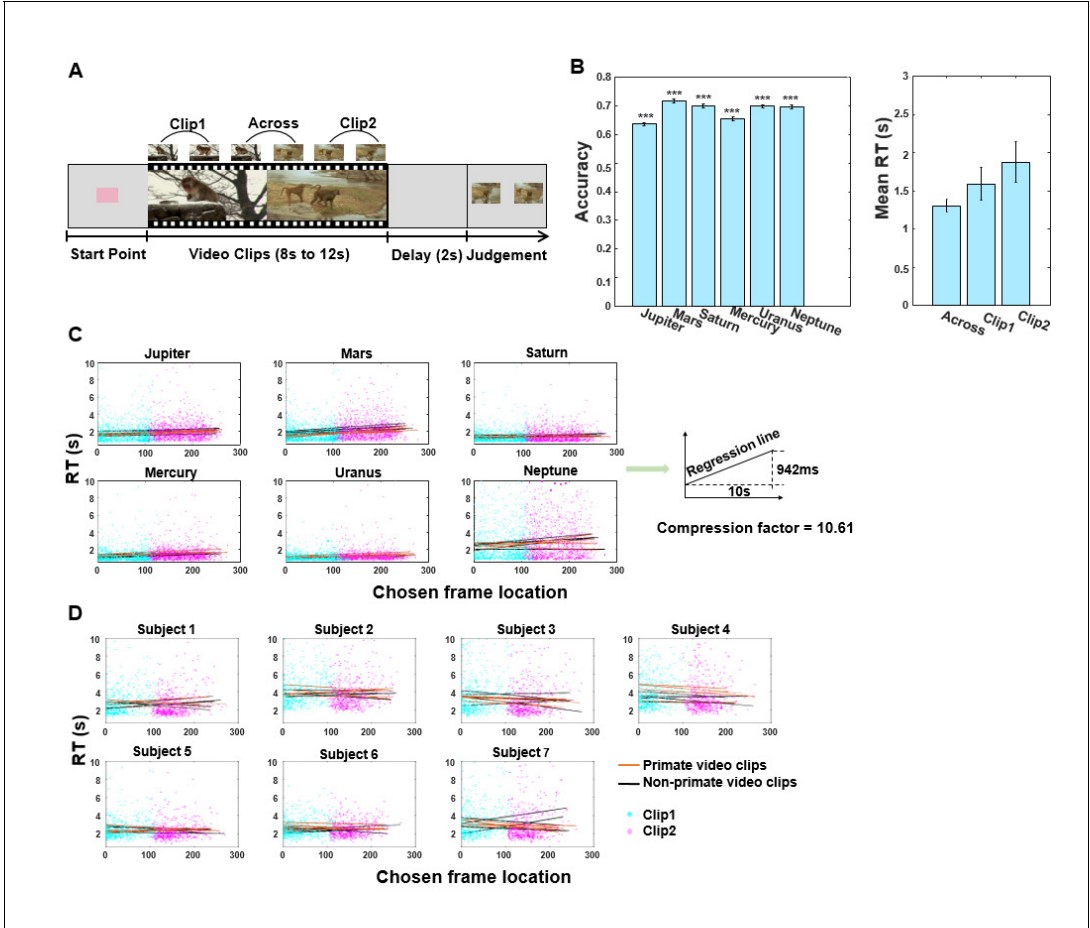

**Figure 1.** TOJ task schema and RT results. (**A**) In each trial, the monkey watched a video (8–12 s, comprising two 4–6 s video clips), and following a 2 s retention delay, made temporal order judgement between two probe frames extracted from the video. The monkeys were required to choose the frame that was presented earlier in the video for water reward. (**B**) Task performance of six monkeys. Proportion correct for the six monkeys (left); mean reaction times for three trial types (right). Error bars are standard errors of the means over monkeys. *** denotes p<0.001. (**C**) Linear plots of reaction time (RT) for each monkey as a function of chosen frame location, see also *Table 1*. (**D**) Linear plots of RT as a function of chosen frame location for each human participant, see also *Supplementary file 4*. In panels (**C**) and (**D**), black lines and orange lines refer to lists of non-primate video clips and primate video clips, respectively (with five repetitions collapsed for monkeys and two repetitions collapsed for human participants). All responses in the within-context condition are shown, with cyan and magenta dots denoting whether the chosen probe frames were extracted from Clip 1 or Clip 2, respectively.

The online version of this article includes the following figure supplement(s) for figure 1:

**Figure supplement 1.** Performance of human participants and speed accuracy trade off results.

used to test for effects arising from context changes (event boundaries), as is discussed in subsequent subsections.

We first addressed our main hypothesis by examining changes in RT using only within-context trials. One possibility is that the monkeys perform TOJ by relying on some form of memory replay, recalling the events coded at specific ordinal positions in a serial manner. To test this hypothesis, in each trial we explicitly looked at the relationship between RT and the location of the frame within the video that the monkeys chose ('chosen frame location', as indexed by the ordinal frame numbers in the video). Considering a range of nuisance variables that might affect these relationships (see also full GLM results in Figure 6), we ran the linear regression analysis of reciprocal latency as a function of chosen frame location/temporal similarity, while including a range of variables as nuisance regressors for each monkey separately. We found a negative relationship between reciprocal latency and chosen frame location in all monkeys (all p<0.001 *; Figure 1C* and *Table 1* upper panel), and between reciprocal latency and temporal similarity (all p<0.001; *Figure 2—figure supplement*

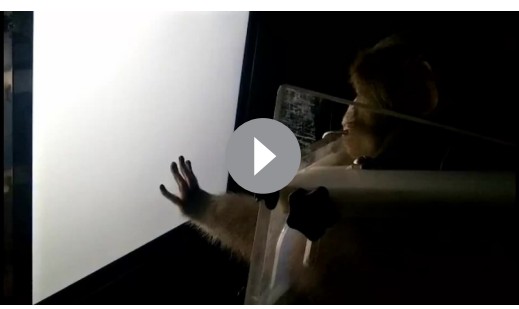

**Video 1.** A monkey performing an example trial. The monkey performs an across-context trial with a correct response (rewarded with liquid). https://elifesciences.org/articles/54519#video1

*1A* and *Supplementary file 2* upper panel). This suggests that the monkeys are systemically faster in identifying frames that are located earlier in the video. Moreover, we replicated these results when considering only correct trials or only incorrect trials separately (*Table 1* middle/bottom panel for chosen frame location; *Supplementary file 2* middle/bottom panels for temporal similarity), suggesting that the putative response latency patterns are not affected by the memory outcome. These patterns of result are also replicated using logarithmically transformed RT data (all p<0.001).

To address the direction and speed of memory replay, reaction times at retrieval were compared between TOJ frames that are within Clip 1 versus Clip 2. During TOJ retrieval, frames that were presented in Clip 1 (mean reaction time = 1.59 s) were retrieved significantly faster than those that were experienced in Clip 2 (mean reaction time = 1.88 s) (one-tailed $t5 = -4.533$; p=0.003; Cohen's d = −0.54; 95% CI: –infinity to −0.158 s; log(RT): one-tailed $t5 = -6.473$; p=$6.558 \times e^{-4}$; Cohen's d = −0.69; 95% CI: –infinity to −0.114 s). Again, these findings confirm that the replay of the video takes place in a forward direction.

Moreover, since the latency required to respond to the chosen frames is much smaller than the duration of the videos themselves, the replay of the video must have been conducted at a compressed speed (i.e., memory replay was faster than perception during video-watching). The difference in reaction time between the very first frame and the last frame was averaged at 942 ms (range: 468–1859 ms). This is equivalent to 94.2 ms to scan through each second of the video, and corresponds to a compression factor of 10.61 during replay in these monkeys (compression factor for each monkey: Jupiter = 13.59, Mars = 7.39, Saturn = 14.80, Mercury = 21.37, Uranus = 17.78, Neptune = 5.38, see *Figure 1C*). This is comparable to a compression factor of 13.7 observed in humans, corroborating the notion of forward replay and those findings in humans (*Michelmann et al., 2019*).

The results suggest that the monkeys' judgments are faster when responding to probe frames that are located in the earlier parts of the videos, and they also illustrate that the search might not follow a linear function. We formally tested for any 'non-linearity' of the fit. We began by testing for a linear relationship and progressively moved to higher-order relationships (quadratic, cubic). At each step, we assessed the significance of the new predictor, having accounted for all the variance fitted by the lower-level predictors. We reported the significance of different trends (linear, quadratic, cubic) for the two species (*Supplementary file 3*). This non-linear (quadratic and cubic) relationship in the monkeys is important because it rules out alternative explanations in which the main effects simply result from positional effects. Rather, the non-linear change in slope indicates that other factors are in play, and that the monkeys might group (or parse) the content according to the relational storyline structure (e.g., the two-clip video design), and do not merely recall the frames/items as of their ordinal positions in a fixed linear manner. In comparison, this analysis also highlights differences between human and macaque monkeys. We found that the human data fits best, and only, with the cubic model (*Supplementary file 3*), suggesting that humans might have treated the video and TOJ differently from the macaques. This difference is reflected in the global compression capability of the human participants (see next section).

In summary, we found that our monkeys might have performed TOJ of video episodes using a forward search of ordered elements in the mnemonic representation at the time of memory test with a non-linear, time-compressed function.

We also ran the same sets of analyses on human participant data for comparison between the two species. Showing a completely opposite pattern, regression analyses on human subjects showed a negative relationship between temporal similarity and reciprocal latency for all participants (all p<0.001, one subject with p=0.055, *Figure 2—figure supplement 1B* and *Supplementary file 4* upper panel). These results imply that the more similar the two frames to be judged, the longer time

**Table 1.** One sample t-tests of the slopes of reciprocal latency as a function of chosen frame location for each monkey after having entered a range of nuisance variables as regressor-of-no-interest (see also *Figure 6*).

The three panels correspond to analyses performed using all trials (top), only correct trials (middle), and only incorrect trials (bottom). The same slope patterns were observed irrespective of response correctness.

| Monkeys | Beta | SEM | t-statistics | p-value | 95% confidence interval Lower | Upper |
|---|---|---|---|---|---|---|
| Slope of reciprocal latency/chosen frame location tested against zero (all trials) | | | | | | |
| Jupiter | –0.203 | 0.021 | –9.751 | <0.001 | –0.244 | –0.163 |
| Mars | –0.369 | 0.025 | –14.950 | <0.001 | –0.417 | –0.320 |
| Saturn | –0.157 | 0.027 | –5.810 | <0.001 | –0.210 | –0.104 |
| Mercury | –0.207 | 0.052 | –3.958 | <0.001 | –0.309 | –0.104 |
| Uranus | –0.164 | 0.022 | –7.595 | <0.001 | –0.207 | –0.122 |
| Neptune | –0.197 | 0.031 | –6.361 | <0.001 | –0.257 | –0.136 |
| Slope of reciprocal latency/chosen frame location tested against zero (correct trials) | | | | | | |
| Jupiter | –0.185 | 0.025 | –7.393 | <0.001 | –0.234 | –0.136 |
| Mars | –0.272 | 0.028 | –9.879 | <0.001 | –0.326 | –0.218 |
| Saturn | –0.092 | 0.032 | –2.857 | 0.004 | –0.155 | –0.029 |
| Mercury | –0.246 | 0.065 | –3.777 | <0.001 | –0.374 | –0.118 |
| Uranus | –0.153 | 0.024 | –6.259 | <0.001 | –0.201 | –0.105 |
| Neptune | –0.150 | 0.039 | –3.858 | <0.001 | –0.226 | –0.074 |
| Slope of reciprocal latency/chosen frame location tested against zero (Incorrect trials) | | | | | | |
| Jupiter | –0.175 | 0.027 | –6.619 | <0.001 | –0.227 | –0.123 |
| Mars | –0.366 | 0.031 | –11.705 | <0.001 | –0.428 | –0.305 |
| Saturn | –0.191 | 0.035 | –5.386 | 0.002 | –0.261 | –0.122 |
| Mercury | –0.075 | 0.077 | –0.975 | 0.330 | –0.227 | 0.076 |
| Uranus | –0.140 | 0.029 | –4.816 | <0.001 | –0.197 | –0.083 |
| Neptune | –0.209 | 0.041 | –5.148 | <0.001 | –0.288 | –0.129 |

needed for retrieve temporal order information. There was also no observable reciprocal latency/ chosen frame location slope in the human data (if anything, it shows an opposite trend; see *Figure 1D* and *Supplementary file 4* bottom panel). It is notable that when reciprocal latency as a function of chosen frame location is analyzed in the humans (as shown in *Figure 2B* compared with *Figure 2A* for monkeys) a very different pattern emerges, suggesting some form of mechanistic discrepancy between the species. We will examine these aspects in detail in the next section.

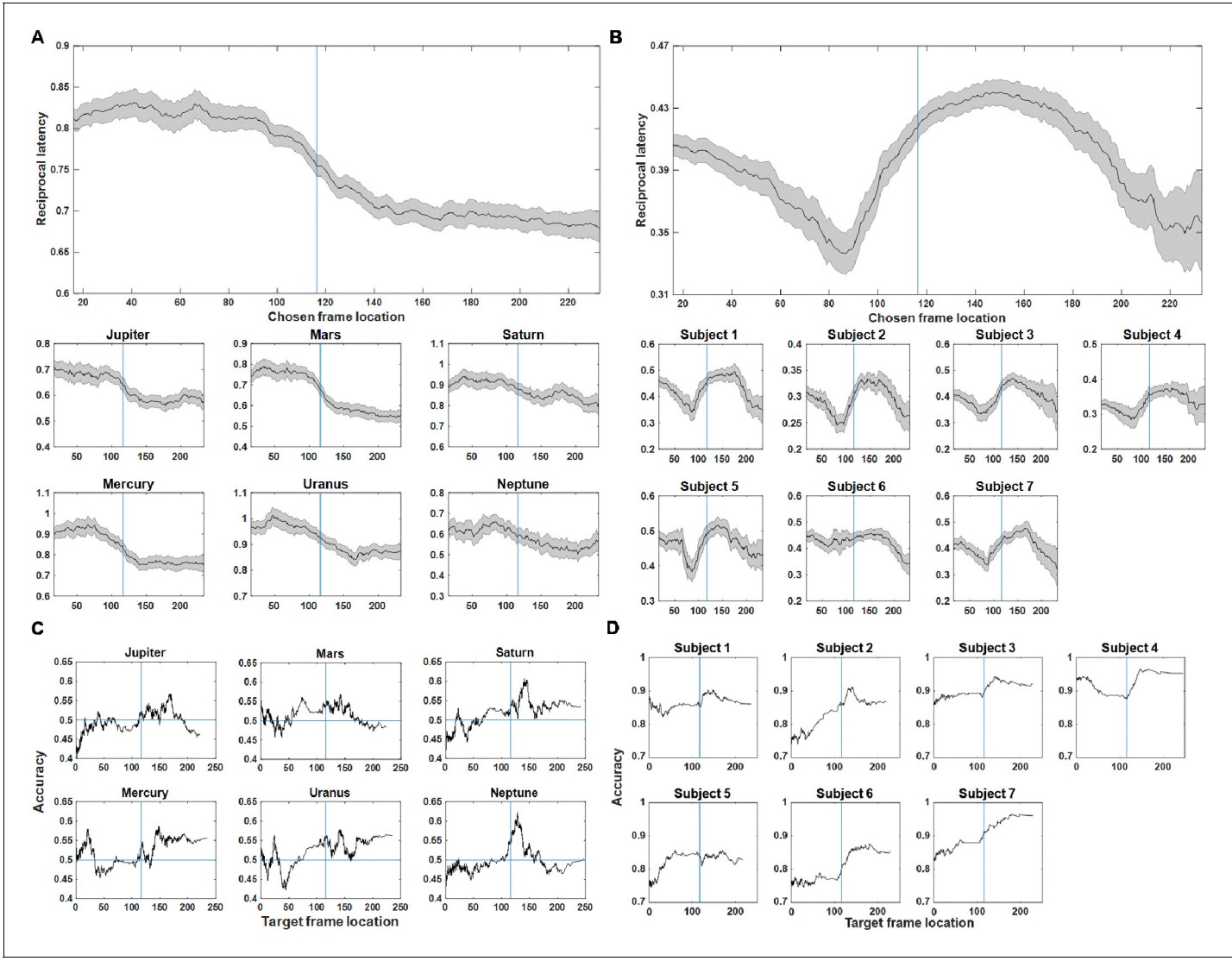

**Figure 2.** Moving average analysis based on reciprocal latency and accuracy for monkeys (left panel) and human participants (right panel). (A) Reciprocal latency for monkeys as a function of chosen frame location for the average of all animals (upper panel) in the within-context condition, with the results for six individual monkeys are shown in the lower panel. The relationship between chosen frame location and RT follows a non-linear pattern. (B) Reciprocal latency for human participants as a function of chosen frame location for the average of all human subjects (upper panel) in the within-context condition, with results for individual subjects are shown in the lower panel. In panels (A) and (B), the shaded region denotes confidence intervals. (C) Proportion of correct answers for individual monkeys as a function of target frame location in the within-context condition. (D) Proportion of correct answers for individual human subjects as a function of target frame location in the within-context condition. In panels (C) and (D), the horizontal blue lines denote chance-level accuracy. Blue vertical lines in these plots denote the mean boundary location between Clip 1 and Clip 2 (116th frame).

The online version of this article includes the following figure supplement(s) for figure 2:

**Figure supplement 1.** Relationship between temporal similarity and reciprocal latency for within-context trials in (A) monkeys and (B) humans.

We also ran a sliding-window average analysis to illustrate how accuracy varies as a function of the target frames' location within within-context trials (*Figure 2C* for monkeys; *Figure 2D* for humans). It is clear that humans can make use of the across-clip boundary to facilitate their TOJ. Interestingly, in paradigms that use discreet images for encoding, the TOJ performance reported in the human literature is distinct from those making use of continuous streaming movie material. Specifically, it has been reported that accuracy and response times are worse in TOJ for items are that separated by a boundary during encoding in the literature (*Ezzyat and Davachi, 2014*; *Heusser et al., 2018*), whereas we show that TOJ performance is better for frames taken from an

across-context condition than for frames taken from a within-context condition in both the proportion of correct answers and RTs (*Figure 1—figure supplement 1A*). This might be because remembering 'discrete segment of events' is facilitated by making use of the contextual or perceptual cues in the video frames in across-context condition. We will pursue this aspect further in a latter part of this manuscript, in which we focus on contrasting within-context trials with across-context trials (results and discussion related to Figure 5).

We also noted that the accuracy might vary along the course of the videos. In order to rule out the possibility that the putative RT effects are not driven by a trade-off in accuracy, we segmented each video into four segments (i.e., each clip was segmented into two segments on the basis of target frame location) and calculated the inverse efficiency score [RT (ms)/percentage correct (%)] for each segment for each individual. The monkeys showed a mild numeral increase in inverse efficiency score across the four segments but this trend did not reach statistical significance (*Figure 1—figure supplement 1B*, left panel) [$F_{(3, 20)} = 0.10$, p=0.96], suggesting that the increase in RT towards the end of the video did not contribute to better accuracy. By contrast, we found that the humans showed a lower inverse efficiency score for the video parts immediately after a boundary (bars 2 and 3 in *Figure 1—figure supplement 1B*, right panel) [$F_{(3, 24)} = 4.17$, p=0.016, a post hoc test shows significant difference between bar 2 and bar 3]. This pattern of boundary effect aligns with the little 'blip' in proportion correct that occurs shortly after the beginning of Clip 2 in the humans (*Figure 2B*). These characteristics might be related to their ability to detect the boundaries.

## Discrepancy with humans: compression of replay is local but not global

It has been shown in the humans that memory replay is not a straightforward recapitulation of the original experience. Subjects can skip through their memories, on a faster time scale across segments of a video episode than within-segment, by skipping flexibly over salient elements such as video boundaries within episodes (*Michelmann et al., 2019*). We propose two possible models with respect to whether the compression is global or not over the whole video. If there is a global compression of the video during replay, the time to initiate replay of Clip 2 would be sooner than the endpoint of replay for Clip 1, as the animal would be able to skip over the whole of Clip 1 to the beginning of Clip 2 (Global-compression model, *Figure 3A*, right panel). However, if the monkeys are not equipped with the ability to skip video segments during the replay process, we would expect a linear increase of retrieval time with chosen frame location, irrespective of the boundary (Strict forward model, *Figure 3A* left). We tested statistically whether the time to initiate replay of Clip 2 was shorter than the duration of Clip 1.

We divided each video into eight equal segments and computed cross-correlations derived from pairs of averaged condition-wise RTs based on chosen frame locations using a representational similarity analysis (see section 'Representational similarity analysis (RSA)' in 'Materials and methods' for details). The RT for TOJ between each segment of the video increases linearly according to their position in encoding (*Figure 3B*, left). We tested these against a hypothetical 'Strict forward' model and found significant correlation with the Strict forward model ($r = 0.66$, p=0.009), but not with the Global compression model ($r = -0.16$, p=0.802) (*Figure 3B*, left). These statistics also remain significant for the Strict forward model when we divided the video into either 10 (p=0.030) or 14 equal segments (p=0.020). The same patterns are also obtained when considering correct trials (Strict forward model: $r = 0.37$, p=0.040; Global compression model: $r = -0.02$, p=0.545) or incorrect trials (Strict forward model: $r = 0.45$, p=0.010; Global compression model: $r = -0.06$, p=0.545) separately. Contrarily, these correlational patterns with the Strict forward model are not observed in the human subjects ($r = -0.11$, p=0.703), but rather we observe a trend favoring the global compression model instead ($r = 0.39$, p=0.069) (*Figure 3B* right). When contrasting the the two models using pairwise comparisons, the two models are both statistically significant for monkeys (p<0.01) and humans (p<0.05), confirming the significance of the winning model.

To control for the confound that the two species might be differentially susceptible to the effect of having different numbers of trials in the experiment, we performed a control analysis that made use of only the first two repetitions of data to equate the number of trials between the two species. With these equated subsets of data, we re-calculated the correlation between monkey RT RDM and the two hypothetical models (i.e., the Strict forward model vs the Global compression model). The results showed a significant correlation with the Strict forward model ($r = 0.56$, p=0.049), but not with the Global compression model ($r = -0.10$, p=0.643) in the monkeys. With this smaller set of

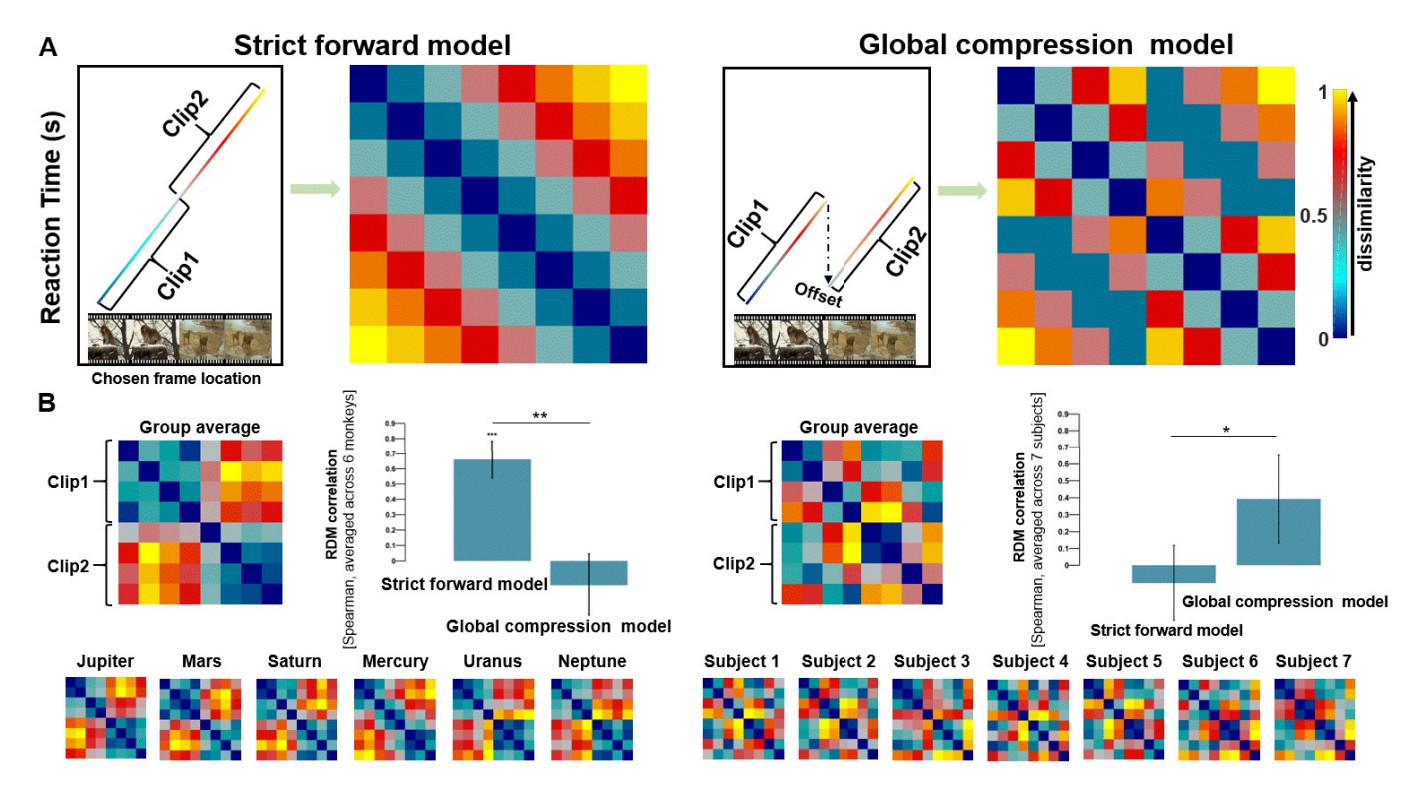

**Figure 3.** Model comparison using representative similarity analysis. (**A**) Visualization of two candidate models as representational dissimilarity matrices (RDMs). Patterns of reaction time (rank-transformed values) as a function of chosen frame location for the the two hypothetical models. The colors of Clip 1 and Clip 2 evolve increasingly with the temporal progression of the video (left), and their respective hypothetical RDMs (right). The reduction in RT (indicated by an arrow) between Clip 1 and Clip 2 is defined as 'offset'; the magnitude of such an 'offset' is arbitrary (but see further analysis in **Figure 4**). (**B**) We segmented the videos into eight equal segments, and the RDMs show pairwise Euclidean distances between these different segments for the species group average (monkeys: left; humans: right) and for each individual separately (monkeys: left bottom; humans: right bottom). RDM correlation tests between behavioral RDMs and two candidate RDMs show that the monkeys replay the footage using a Strict forward strategy ($r = 0.66$, p=0.009), and provide little evidence for the Global compression strategy ($r = -0.16$, p=0.802). Humans show an opposite pattern from the macaques. In the humans, the Global compression model shows a higher correlation with behavioral RDM (marginally insignificant $r = 0.39$, p=0.069) than with the Strict forward model ($r = -0.11$, p=0.703). Pairwise comparisons show that the two models are both statistically significant for monkeys (p<0.01) and humans (p<0.05). Error bars indicate the standard errors of the means based on 100 iterations of randomization. *P* values are FDR-corrected (***, p<0.001, **, p<0.01, *, p<0.05).

data, we also directly compared the correlations between the two models for the monkeys and found significant differences between them (p<0.05, FDR-corrected). These results replicated the findings observed when the full set of data were used, suggesting that more numerous exposures to the same stimuli did not affect the main results.

## Factors modulating the model: 'offsets' for search and memory-search RT slope

We defined the reduced RT to initiate replay for Clip 2 as 'offsets' in initiating search in Clip 2 by skipping the non-informative Clip 1 (**Figure 3A**). With respect to the detailed differences between the two models, one may wonder whether and how the 'offsets' between Clip 1 and Clip 2 might influence the results. Especially for the Global compression model, changes of this parameter will cause changes in the RDMs. To address this concern, we simulated an array of RDMs by systemically varying the offset parameter and produced 11 hypothetical models, ranging from an absolute Global compression model (model 1, most left in **Figure 4A**) to a Strict-forward model (model 6, middle in **Figure 4A**), and beyond (7th to 11th models, right in **Figure 4A**). We then tested each individual monkeys' data with each of these 11 models. The results show that the Spearman correlation values

between the monkey's data and hypothetical RDMs reach an asymptote of around $r = 0.8$ as the offset parameter tends to zero, and notably, that the correlation values only improve minimally with increasing offsets (*Figure 4B*, with each individual's RT RDM displayed as an inset). These results suggest that the monkeys have processed the video as a holistic chunk of information rather than taking advantage of skipping the non-informative first clip when the two probe frames were in Clip 2. For comparison, we also tested human participant's data against each of these 11 hypothetical models and found a completely opposite pattern in the humans (*Figure 4C*, with each individual's RT RDM displayed as an inset). Taken together, we reveal a discrepancy between human and macaque performance in terms of their ability to compress past (irrelevant) information during TOJ.

## Context changes (event boundaries) increase the rate of rise in decision information

Thus far, we have focused on how the monkeys retrieve the order of frames when information was equated within contexts, but how contextual changes might aid TOJ processes remains to be examined. It was evident that the monkeys retrieved the temporal order of frames with numerally different speeds for the three trial-types: across-Clip 1 and Clip 2 vs. within-Clip1 vs. within-Clip2 [$F_{(2, 15)} = 2.32$, p=0.132 (*Figure 1B*, right)]. Thus, we then compared the latency distribution of within-context and across-context conditions. and we hypothesized that a context shift would change the rate of rise of information accumulation (shift model) without altering the decision threshold (swivel model) within the Linear Approach to Threshold with Ergodic Rate (LATER) model (*Figure 5A*). We compared across-context and within-context trials specifically and fitted the two types of LATER models to each monkey's data separately [*Figure 5B*, see section 'LATER (linear approach to threshold with ergodic rate) modelling' in 'Materials and methods'], together with a 'two fits' model, which supposes that the reaction times for the two conditions are independent of each other, and a null model, which assumes that there is no effect of manipulation. Using the Bayesian information criterion (BIC) as an index of model comparison, the results consistently indicate that the shift model is better than the swivel model for all six monkeys [range of $\Delta$BIC = (14.57, 300.07); *Supplementary file 3*, see section 'Model comparison' in 'Materials and methods'). These results further indicate that contextual changes do not alter the judgement threshold for decisions (providing no evidence for a swivel pattern). By contrast, this pattern is not seen in the human participants (*Figure 5C* and *Supplementary file 3*), suggesting that the two species might not treat the information given by the event boundary during TOJ in the same manner. Within a drift diffusion model framework, the results suggest that monkeys accumulate information for memory decisions at a faster rate when the frames were extracted from two different clips than when the frames were extracted from the same-context clip.

## Confirmatory GLMs and control analyses for the putative patterns

To verify whether the effects are attributed to basic stimulus features such as the perceptual differences inherent in the across-context condition. We then generated several generalized linear models to quantify the effect sizes of several principal variables (see section 'Generalized linear models (GLM)' in 'Materials and methods'). In the within-context condition, given that the monkeys would replay their experience to judge the relative temporal order of probe frames ('replay hypothesis'), we used the temporal characteristics of probe frames, as represented by chosen frame location (or temporal similarity, which is essentially an inverse of frame location) as the independent variables. In the across-context condition, we included a perceptual similarity measure, which was based on feature points extracted by the SURF algorithm (SURF similarity, see *Figure 6—figure supplement 1C*), in the GLM to reflect the extent to which the monkeys were able to capitalize on using contextual boundaries for TOJ judgment. In addition, we also entered a number of independent variables as regressors: a binary regressor indicating whether the video includes primate content or not, a binary regressor indicating that a video is played forward or backward, the five repetitions (or two repetitions for humans) of the video-trials, physical location of the selected probe (left or right), time elapsed within a session, chosen frame location, temporal similarity, perceptual similarity (SURF), temporal distance, and response of the subjects (correct/incorrect).

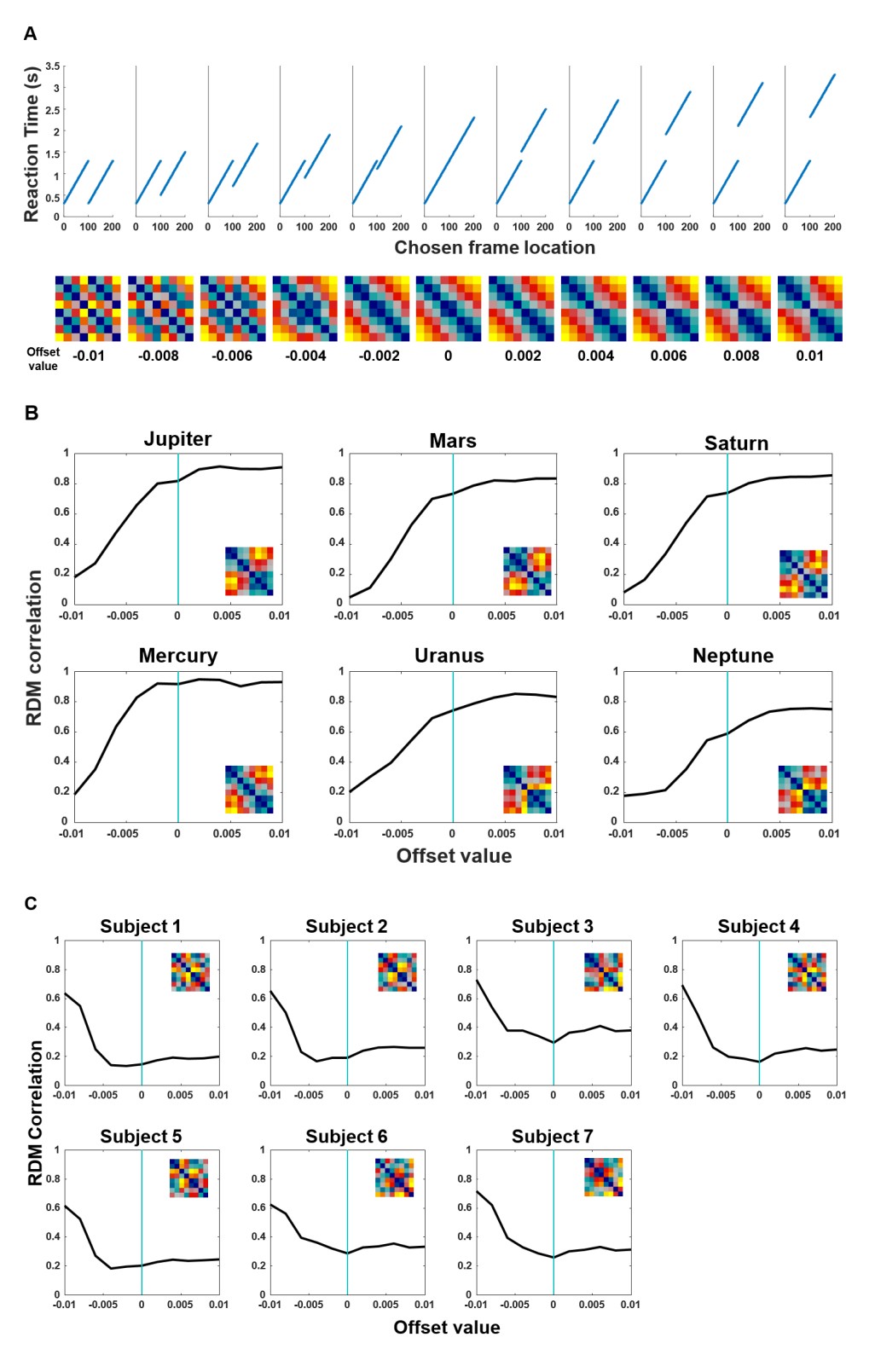

**Figure 4.** The Strict forward model provides a better fit to the RT data in monkeys but not in humans. (**A**) 'Offsets' are defined as the magnitude of reduced RT when the frames were in Clip 2. 11 hypothetical models with their reaction time patterns (top) and RDMs (bottom). We systemically varied the 'offset' parameter while keeping a constant slope. These models progressively range from an absolute Global compression model (model 1, most left) to a Strict-forward model (model 6, middle), and beyond (7th to 11th models, right). The numerals below the RDMs denote the magnitude of the

*Figure 4 continued on next page*

*Figure 4 continued*

respective offsets. (B) Each monkey's data were tested against each of these 11 hypothetical models. The Spearman correlations increase as a function of offset magnitude between Clip 1 and Clip 2 until reaching an asymptote when the offset value is around zero, which corresponds to the Strict forward model (model 6 in panel (A); see also *Figure 3*). Individuals' RT RDMs are shown in insets. (C) Each human participant's data were also tested against each of these 11 hypothetical models. The Spearman correlations decrease as a function of offset magnitude between Clip 1 and Clip 2 until reaching an asymptote when the offset value is around zero. This analysis confirms the hypothetical discrepancy between the two species (see also *Figure 3B*).

The within-context GLM shows that monkeys' RT was indeed significantly faster when the probe frame was located earlier in the video, p<0.001 (or in equivalent terms, when two frames were temporally closer, p=0.004), confirming our main finding that the monkeys might have adopted a forward scanning strategy for information retrieval (*Figure 6A*, left). By contrast, the across-context GLM shows that there was no significant effect of the chosen frame location on RT. Rather, the monkeys retrieve their memories significantly faster for probe frames that are contextually (or perceptually) distinct, p=0.004 (*Figure 6A* , right). Human results are shown in *Figure 6B* for comparison.

Finally, considering that the monkeys performed a larger number of trials than the human participants (5000 vs. 2000 trials), we plotted the accuracy data for each testing day for each monkey (and for humans too, *Figure 7*) and observed no apparent increase in performance over the course of the 50 testing days. This absence of performance change could also be due to the extensive training that the monkeys had received prior to this experiment (~21,150 trials in total for each monkey across 141 sessions on average, see *Table 2*). We therefore deem the main effects reported here as unlikely to be attributable to behavioral or strategic changes over the course of main experiment.

## Discussion

In light of recent reports on the neural correlates underlying how humans and rodents replay their past experiences (*Liu et al., 2019*; *Michelmann et al., 2019*; *Panoz-Brown et al., 2018*; *Davidson et al., 2009*), here, we demonstrate parallel behavioral findings in macaque monkeys looking at dynamic cinematic material. Previous reports of macaques succeeding in TOJ indicated their ability to remember the order of events (*Templer and Hampton, 2013*; *Martin-Ordas et al., 2010*; *Ninokura et al., 2003*) and even to monitor the quality of representations of temporal relations among item images meta-cognitively (*Templer et al., 2018*). For example, *Orlov et al., 2000* suggested that monkeys can categorize stimuli by their ordinal number to aid recall of order, and *Templer and Hampton, 2013* showed that monkeys retrieve the temporal order information on the basis of the order of events rather than elapsed time.

One possible common mechanism underlying these performances is that monkeys use a forward search to identify targets in memory representation (*Gower, 1992*). Taking advantage of the latency data obtained during TOJ on naturalistic materials, we provide new behavioral evidence in support of the hypothesis proposed by *Gower, 1992* that the monkeys can replay their memory in a serial forward manner. Our analysis further clarifies that this replay process is conducted in a time-compressed manner. Notwithstanding task differences, both humans and macaques execute retrieval with forward replay with a comparable compression factor (factors of ~11 in macaques vs. ~13 in humans, *Michelmann et al., 2019*) (but see also *Wimmer et al., 2019*). Another cross-species similarity rests on the observation that the RT patterns are independent of retrieval success. We have previously showed in analogous TOJ paradigms in humans that TOJ task-specific BOLD activation and behavioral RT patterns are independent of retrieval accuracy (*Kwok et al., 2012*; *Kwok and Macaluso, 2015a*). We interpreted these effects as process-based rather than content-based. At present, the latency results probably also point towards some 'search' or replay processes, and any ultimate incorrect responses are thus likely to be caused by memory noises injected during encoding and/or during delay maintenance.

Despite the cross-species similarity, our revelation of a critical discrepancy between humans and macaques carries an important theoretical implication: humans can do both local and global compression, whereas monkeys are not able to attain global compression. The implication is that mental time travel is not all-or-none. There could be multiple layers underpinning the concept of mental time travel, which entail the ability to relive the past (*Suddendorf et al., 2009*; *Tulving, 1985*) and

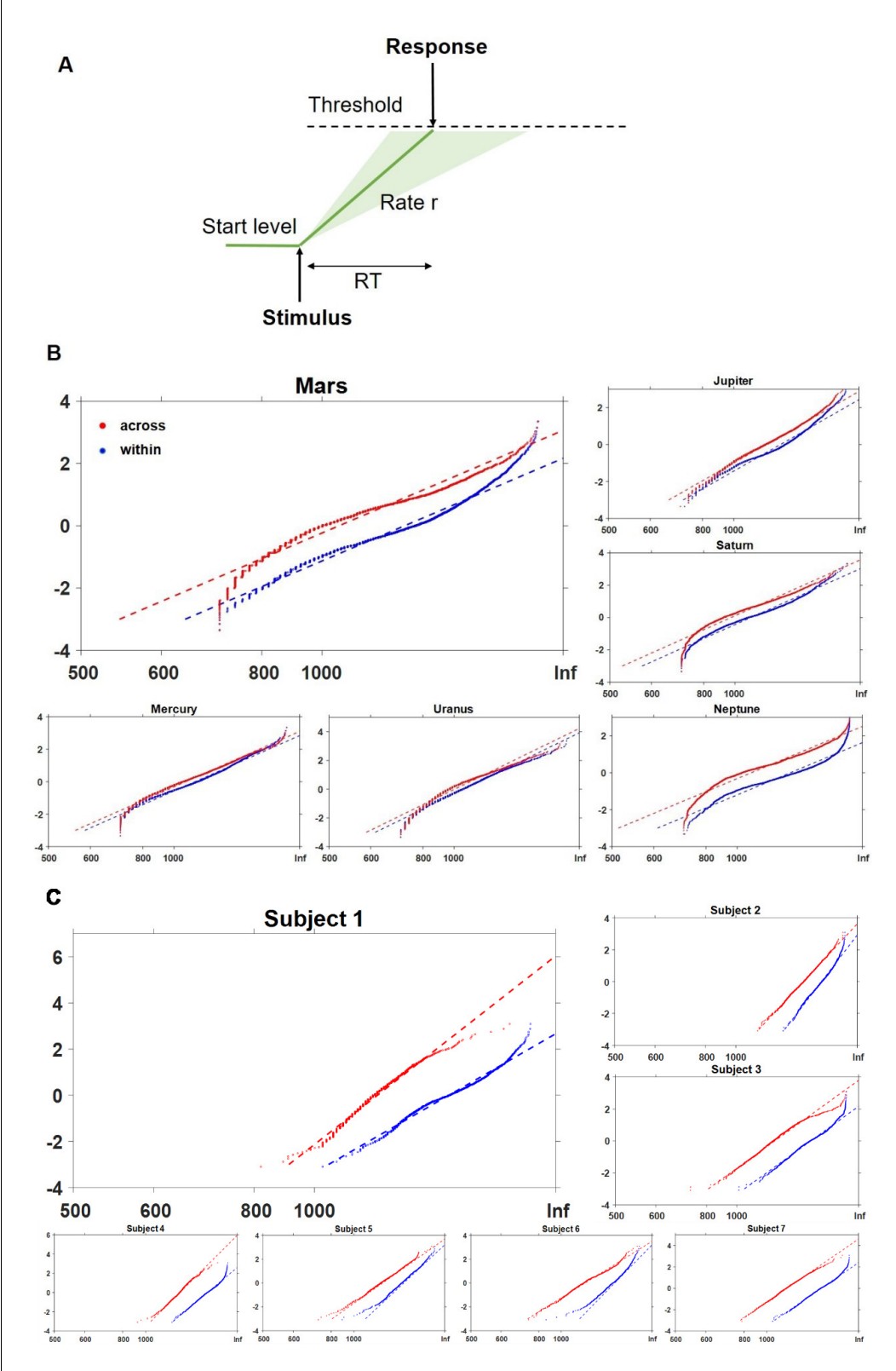

**Figure 5.** LATER model fitting of RT in across-context and within-context conditions for both species. (A) Cartoon of the LATER model cartoon illustrating that a decision signal triggered by a stimulus rises from its start level, at a rate of information accumulation *r*, to the threshold. Once it reaches the threshold, a decision is initiated. The rate of rise *r* varies from trial to trial, obeying the Gaussian distribution (variation denoted as green shaded area). (B) Contextual change effect on the distribution of response latency for the monkeys; data from Monkey 'Mars' was chosen for larger

*Figure 5 continued on next page*

*Figure 5 continued*

display. (C) Contextual change effect on the distribution of response latency for humans; data from Subject 1 was chosen for larger display. The red and blue dashed lines show the best fits (maximum likelihood) of across-context trials and within-context trials, respectively (see also *Supplementary file 3*).

skipping over unimportant information (*Michelmann et al., 2019*). The latter aspect allows humans to recall our memories flexibly far into our past and suggest powerful computational efficiencies that may facilitate memory storage and recall. By contrast, we did not find any evidence in the macaques of an ability to make use of salient boundary cues to skip unimportant details. It thus remains unknown how far back in time monkeys can replay their memories. It has been shown recently that humans can spontaneously replay experience on the basis of learned structures, with fast structural generalization to new experiences facilitated by representing structural information in a format that is independent of its sensory consequences (*Liu et al., 2019*). The lack of global compression in the monkeys of their video experience implies that the monkeys might not be able to use factorized representations to allow components to be recombined in more ways than were experienced (*Behrens et al., 2018*). However, by establishing that neither the number of intervening frames nor the passage of time per se determines the RT pattern (probably a mixed effect resultant from a combination of both), we ruled out order or positional memory as the underlying mechanism supporting TOJ in this task.

Although we have argued for the presence of replay-like patterns in the monkey during TOJ, we are aware that replay is a neural phenomenon supported by the activity of individual neurons and that implicates the offline reactivation of sequences of hippocampal place cells that reflect past and future trajectories (*Pfeiffer and Foster, 2013*; *Jadhav et al., 2012*; *Ólafsdóttir et al., 2018*). On the basis of the behavioral data presented here, the question of where the putative replay may be occurring anatomically is important for the broader field. On the one hand, in rodent research, replay has been linked to sharp wave-ripples in the hippocampal formation (*Lee and Wilson, 2002*; *Foster and Wilson, 2006*). On the other hand, recent MEG studies on humans' replay have implicated several cortical areas such as the occipito-parietal cortex (*Michelmann et al., 2016*), the vmPFC (*Liu et al., 2019*), and the visual cortex (*Wimmer et al., 2019*), in addition to the MTL including the hippocampus (*Liu et al., 2019*; *Wimmer et al., 2019*). These observations, made under simultaneous whole-brain recordings in humans, are in line with the idea that replay may be coordinated between the hippocampus and neocortical areas (*Ji and Wilson, 2007*).

A further caveat is that although most of the rodents studies on replay focus on spontaneous replay patterns at rest (*Liu et al., 2019*), during sleep (*Lee and Wilson, 2002*; *Louie and Wilson, 2001*) or during a task-free state (*Foster and Wilson, 2006*; *Karlsson and Frank, 2009*), our monkeys perform their replay-like recall of the videos as an effortful operation to solve a TOJ task. This is more akin to studies using stimuli embedded in episodes (*Wimmer et al., 2019*) or short video-episodes (*Michelmann et al., 2016*) for their ecological validity. Our combined results thus constitute a novel connection between various kinds of replay-like behaviors that are shared between rodents and humans, and provide a primate model for anatomical investigation. This cognitive discrepancy should be further elucidated using electrophysiological methods probing into the MTL (*Davidson et al., 2009*; *Foster and Wilson, 2006*; *Karlsson and Frank, 2009*; *Diba and Buzsáki, 2007*) and the neocortices (*Naya et al., 2017*).

We observed one further interesting phenomenon here, which is that the monkeys are able to detect contextual changes to facilitate TOJ. We show that the expedited TOJ in across-context condition was facilitated by contextual details, which in turn results in an increased rate of rise of signal towards memory decision in a drift-diffusion process. Humans studies show that contextual changes lead to segmentation of ongoing information (*Magliano et al., 2001*; *Hard et al., 2006*). Our results provide evidence consistent with event segmentation in the macaque monkeys and imply that these monkeys might be capable of parsing the footage using contextual information, akin to what has been shown in humans (*DuBrow and Davachi, 2013*; *Sols et al., 2017*; *Ezzyat and Davachi, 2011*; *Kwok and Macaluso, 2015b*) and rodents (*Panoz-Brown et al., 2016*).

Memory replay is an elaborate mental process and our demonstration of a time-compressed, forward replay-like pattern in the macaque monkeys, together with their primordial rigidity in compressing the experienced past, provides promise for mapping the evolution of episodic memory in our lineage.

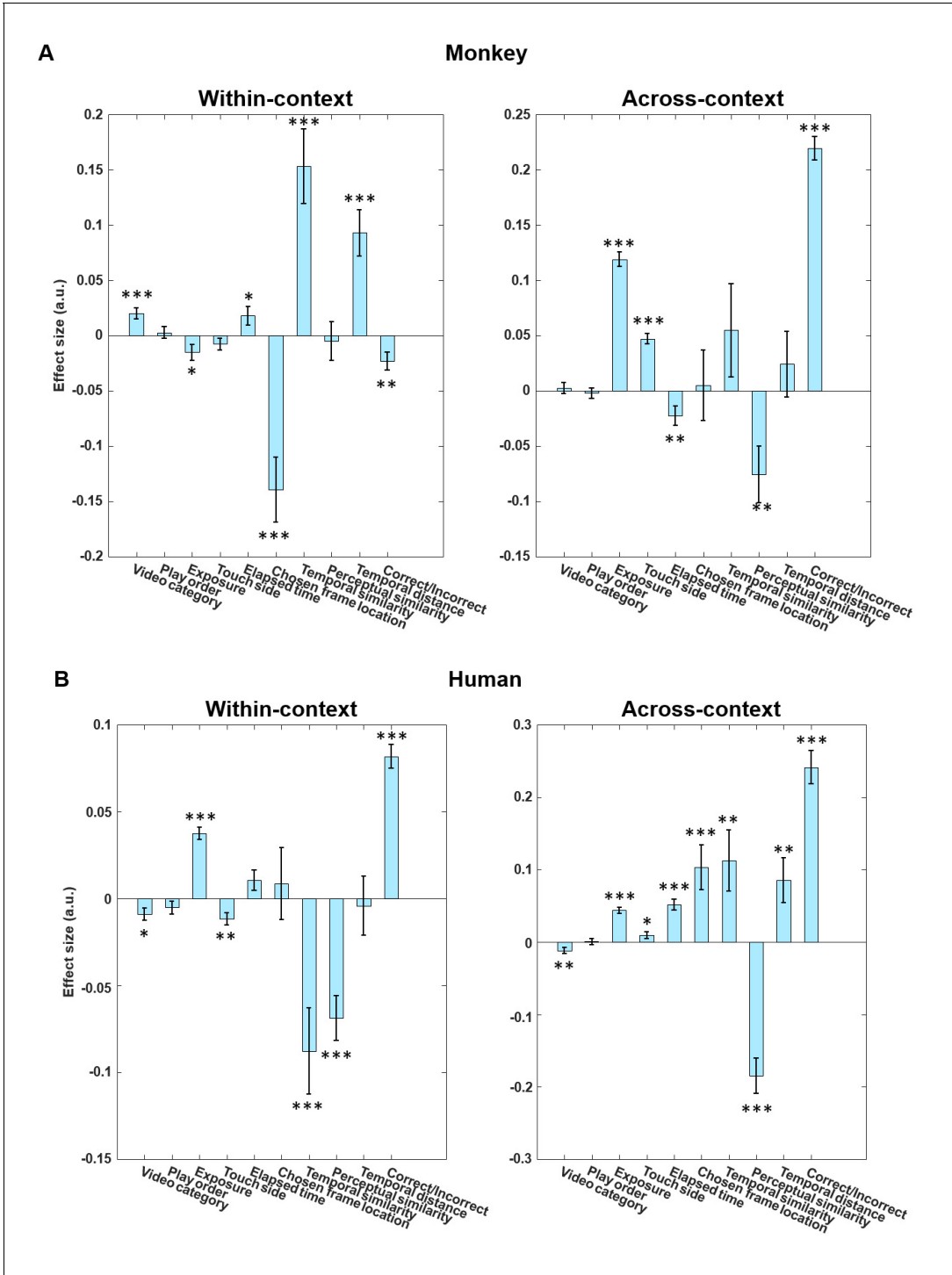

**Figure 6.** Full GLM analysis including a number of variables that might affect reciprocal latency separately for within-context and across-context conditions. (**A**) Monkey data. (**B**) Human data. We included ten regressors, namely, a binary regressor indicating whether the video category is primate or non-primate (video category), a binary regressor indicating that a video is played forward or backward (play order), the repeated exposure of the trial (Monkey: 1–5; Human: 1–2) (exposure), the physical location of the selected probe on screen (left or right) (touch side), time elapsed within a session (elapsed time; to rule out fatigue or attentional confounds), chosen frame location, temporal similarity, SURF similarity as a perceptual similarity measure (perceptual similarity), temporal distance between two probe frames, and response of subjects (correct/incorrect). In the monkeys, the results confirm that chosen frame location is the most significant regressor in within-context trials, whereas perceptual similarity is the most significant regressor in across-context trials. ***, p<0.001, **, p<0.01, *, p<0.05.

The online version of this article includes the following figure supplement(s) for figure 6:

*Figure 6 continued on next page*

*Figure 6 continued*

**Figure supplement 1.** GLM results on the effects of image similarity measures on reciprocal latency for within-context and across-context conditions for (A) the monkeys and (B) humans, and (C) an example of SURF similarity.

## Materials and methods

### Subjects

Macaque monkeys

Six male rhesus macaques (*Macaca mulatta*) (5.49 ± 0.5 kg) with a mean age of 3.5 years at the start of testing participated in this study. They were initially housed in a group of 6 in a specially built spacious enclosure (max capacity = 12–16 adults) with enrichment elements such as a swing and climbing structures present until the study began. The monkeys were then housed in pairs during the experimentation period according to their social hierarchy and temperament. They were fed twice a day with portions of 180 g monkey chow and pieces of fruits (8:30am/4:00pm). Water was available ad libitum except on experimental days. They were routinely offered treats such as peanuts, raisins and various kinds of seeds in their home cage for forage purpose. The monkeys were procured from a nationally accredited colony located in the Beijing outskirts, where the monkeys were bred and reared. The animals were thus ecologically naive to the natural wilderness and should not have had any previous encounter with other creatures except humans and their companion. The room in which they are housed is operated with an automated 12:12 (7am/7pm) light-dark cycle and kept within temperate around 18–23°C and humidity of 60–80%.

Human subjects

Seven participants (mean age = 19.57 ± 1.13, 6 female) took part in the experiment. The participants were recruited from the undergraduate population in East China Normal University. The participants provided informed consent and were compensated 400 RMB for their time.

The experimental protocol was approved by the Institutional Animal Care and Use Committee (permission code: M020150902 and M020150902-2018) and the University Committee on Human Research Protection (permission code: HR 023–2017) at East China Normal University. All experimental protocols and animal welfare adhered with the 'NIH Guidelines for the Care and Use of Laboratory Animals'.

### Training history and task performance

There were five stages of training on the TOJ task and the numbers of days per monkey are reported in *Table 2*. With these extended periods of training, the monkeys' performances were unlikely to change over the course of the main experiment (see also *Figure 7*).

### Apparatus and testing cubicle

Macaque monkeys

The testing was conducted in an automated test apparatus controlled by two Windows computers (OptiPlex 3020, Dell). The subject sat, head-unrestrained, in a wheeled, specially made Plexiglas monkey chair (29.4 cm × 30.8 cm × 55 cm) fixed in position in front of a 17-inch infrared touch-sensitive screen (An-210W02CM, Shenzhen Anmite Technology Co., Ltd, China) with a refresh rate of 60 Hz. The distance between the subject's head and the screen was kept at ~20 cm. The touch-sensitive screen was mounted firmly on a custom-made metal frame (18.5 cm × 53.2 cm) on a large platform (100 cm × 150 cm × 76 cm). Water reward delivery was controlled by an automated water-delivery rewarding system (5-RLD-D1, Crist Instrument Co., Inc, U.S.) and each delivery was accompanied by an audible click. An infrared camera and video recording system (EZVIZ-C2C, Hangzhou Ezviz Network Co., Ltd, China) allowed the subject to be monitored while it was engaged in the task. The entire apparatus was housed in a sound-proof experimental cubicle that was dark apart from the background illumination from the touch screen.

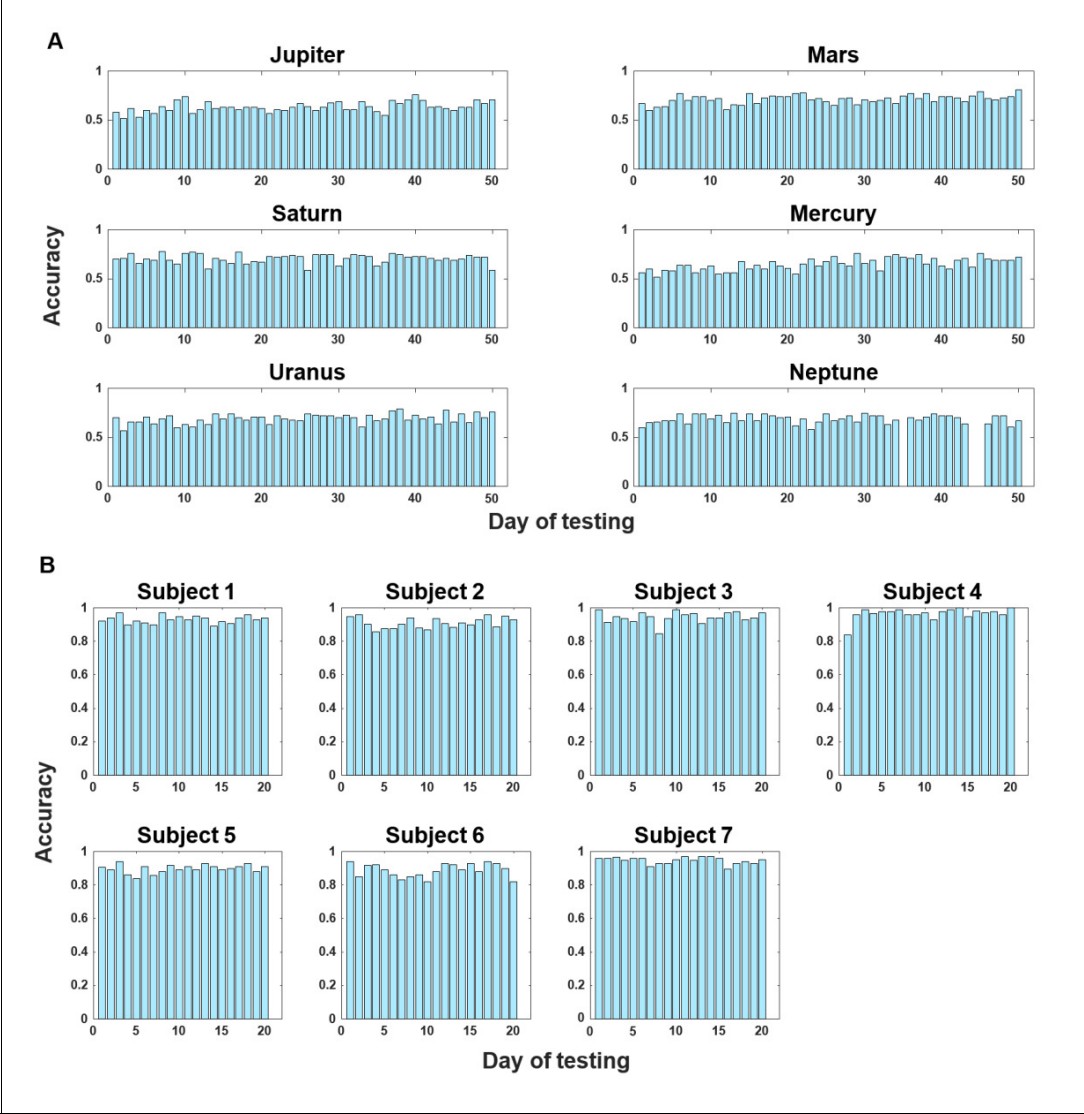

**Figure 7.** Sessional accuracy data expressed as proportion correct for each individual. (A) Monkey data. (B) Human data. No obvious increase in performance was observed over the course of testing days in the experiment for either monkeys or humans.

**Table 2.** Number of days accrued in each training stage.

When we trained the two additional monkeys (Uranus and Neptune), we made them skip the 4th and 5th stages entirely. The performance patterns of these two monkeys were not different from those of the initial four.

| Monkeys | Stage 1 | Stage 2 | Stage 3 | Stage 4 | Stage 5 |
|---------|---------|---------|---------|---------|---------|
| Jupiter | 12 | 39 | 16 | 65 | 51 |
| Mars | 29 | 7 | 15 | 48 | 42 |
| Saturn | 25 | 42 | 31 | 64 | 95 |
| Mercury | 13 | 38 | 17 | 61 | 49 |
| Uranus | 14 | 17 | 11 | - | - |
| Neptune | 13 | 21 | 11 | - | - |

## Human subjects

We used the same model of computers and touch-sensitive screens for the human participants. The human subjects sat ~30 cm in front of another identical 17-inch infrared touch-sensitive screen, with stimuli presented with the same computer used in the monkeys' experiment.

## Source of video materials and preparation

A collection of documentary films on wild animals was gathered from YouTube. The films were *Monkey Kingdom* (Disney), *Monkey Planet* (Episode 1–3; BBC), *Planet Earth* (Episode 1–11; BBC), *Life* (Episode 1–10; BBC), and *Snow Monkey* (PBS Nature). In total, 28 hr of footage was gathered. We applied Video Studio X8 (Core Corporation) to parse the footage into smaller segments. Experimenters then applied the following criteria to edit out ~2500 unique clips manually: 1) the clip must contain a continuous flow of depiction of events (i.e., no scene transition); 2) at least one living creature must be included; 3a) at least one of the animals contained must be in obvious motion; 3b) the trajectories of these motions must be unidirectional (i.e., no back and forth motion of the same subject); 4) clips with snakes were discarded. From this library, we then selected 2000 clips (all of 4 s – 6 s) for the final test, and a small number of additional clips were also prepared for the training stages. Each of the monkeys was assigned with a pseudo-randomized remix of a unique 1000 video-trials set from the 2000-video library, thus eliminating video idiosyncrasy associated with any experimental conditions. For each individual monkey, the videos assigned to be in an experimental condition were not used/shown in another condition, so that a particular monkey would only view that video repeatedly under the same condition.

## Task and experimental procedure

We combined naturalistic material with a TOJ paradigm that is widely used in episodic memory research (*Templer and Hampton, 2013*; *Manns et al., 2007*; *Kwok and Macaluso, 2015c*). In each trial, the monkey initiated a trial by pressing a colored rectangle in the center of the screen (0.15 ml water). An 8–12 s video (consisting of two 4–6 s clips) was then presented (0.15 ml water), and following a 2 s retention delay, two frames extracted from the video were displayed bilaterally on the screen for TOJ. The monkeys were trained to choose the frame that was shown earlier in the video (see *Video 1*). A touch to the target frame resulted in 1.5 ml water as reward, the incorrect frame was removed, and the target frame remained alone for 5 s as positive feedback. A touch to the incorrect frame removed both frames from the screen and blanked the screen for 20 s without water delivery. As the monkeys could self-start the trials, we did not set an explicit inter-trial interval. Correction trial procedures were not used in the main test.

We collected 50 daily sessions of data. Each session contained 100 trials, giving us a 5000 trials per monkey. The 5000 trials contained a break-down of four factors: Boundary (Within vs. Across), Play Order (Normal vs. Reverse), Temporal distance (TD, 25 levels), and Exposure (R1–R5), giving out a 2 * 2 * 25 * 5 within-subject design. The two frames for TOJ could be extracted from the same clip (Within) or from distinct clips (Across). TD was an ordinal variable, with 25 levels ranging between a minimum of 1000 ms (equivalent to 25 frames) and a maximum of 3880 ms (equivalent of to frames). Each TD level increased progressively with three frames in each step. The ten lists contained five lists of primate animals and five lists of non-primate animals (Category: Primate/Non-Primate). The six monkeys were counter-balanced in their order of receiving the two kinds of material: three monkeys (Jupiter, Mars, and Saturn) were tested first on Non-Primate lists, whereas the other three (Mercury, Neptune, and Uranus) were tested first on Primate lists. The whole experiment lasted for 68 days. There were three blocks of reset (3 days, 9 days and 6 days) in-between testing days. The task was programmed in PsychoPy2 implemented in Python.

The same experimental design was adopted by the human participants. They re-used the 6 unique video-trials/list sets and TOJ frames (one unique set per monkey) correspondingly for the human subjects (the extra subject 7 re-used set 1). The only critical difference was that human participants performed only two exposures (R1–R2) rather than five. Identically, each session contained 100 trials, and with 20 daily sessions of testing, we acquired 2000 trials per participant for analysis.

## Data analysis: modelling, temporal and perceptual similarity, and model comparison

Trials with RT longer than 10 s (1.45%) or shorter than 0.7 s (2.47%) were excluded from the analyses. Both correct and incorrect trials were entered for all analyses. For Neptune, data from one day of primate list 5 (exposure 4) and from two days of primate lists 4 and 5 (exposure 5) were lost because of machine breakdown and thus only 4700 trials were included for Neptune.

### Representational similarity analysis (RSA)

To create RDMs for the representational pattern of response time, we evenly divided each video into eight segments on the basis of chosen frame location and averaged the response times within each segment for each monkey individually. For each matrix, we then computed the Euclidean distance of average response time for each pair of the segments. In order to show the relative distance among all pairs of segments intuitively, we further transformed the matrix by replacing each element with the rank number in the distribution of all the elements, and linearly scaled into [0,1]. To compare similarity between RT patterns and the candidate models, we computed Spearman correlation between the model and group averaged RT RDMs with 100 randomized iterations by bootstrapping. We then used two-sided Wilcoxon signed-rank test to test the null hypothesis that the correlation between data RDM and the two hypothetical models are equal. The statistical threshold was set at $p<0.05$ (FDR corrected).

### LATER (linear approach to threshold with ergodic rate) modelling

The observations that the brain needs more time than that requires for nerves to transport information and that trial-by-trial RTs vary considerably have stimulated researchers to make use of distribution of RT to examine mental processes (*Magliano et al., 2001*). Latency, as an indicator of decision processes, provides a source of insight into the underlying decision mechanisms (*Louie and Wilson, 2001*; *Karlsson and Frank, 2009*; *Diba and Buzsáki, 2007*). The Linear Approach to Threshold with Ergodic Rate (LATER) model is a widely used model that taps into these processes (*Noorani and Carpenter, 2016*). The LATER model stipulates that the winner signals that reach threshold faster would trigger the decision, resulting in shorter latency. Accordingly, changing the rate of rise would cause the line to 'shift' along the abscissa without changing its slope. By contrast, lines plotted by latency distribution would 'swivel' around an intercept when the threshold changes (*Reddi and Carpenter, 2000*; *Figure 5A*). To explore the underlying mechanisms of different experiment conditions, we plotted the reciprocal of RT (i.e., a multiplicative inverse of RT, 1/RT) as a function of their z-scores, thus making the distributions follow a Gaussian distribution (*Noorani and Carpenter, 2011*). Hence, we fitted the main component for each condition. The main component is a ramp to the threshold with rate of rise $r$. The distance between start level $S_0$ and threshold $S_T$ is defined as $\theta$, and the rate of rise $r$ follows a Gaussian distribution of mean, $\mu$, and standard deviation, $\sigma 1$.

For model comparison, we fitted four different models to the data. The 'null' model fits the RT from within-context vs. across-context conditions with the same parameters, implying no effect of manipulation. The 'two fits' model set all the parameters to be free, which supposes that the RTs of the two conditions are independent of each other. The 'Shift' model only allows the slope of main component $\mu$ to change according to different conditions on the assumption that the manipulation will change the rate of rise. The 'Swivel' model only allows $\theta$ to change according to different conditions on the assumption that the subject sets different thresholds for different conditions.

### Generalized linear models (GLM)

We ran GLM to compare the effect sizes of independent variables on the dependent variable. The mean ($\mu$) of the outcome distribution $Y$ depends on the independent variables $X$, according to the following formula:

$$\mathrm{E}(Y) = \mu = g^{-1}(X\beta)$$

where $Y$ is a distribution of outcomes, $\beta$ is an unknown parameter to be estimated, and $g$ is a link function (Gaussian function). The dependent variable (Y) is reciprocal latency, and the independent variables are as follows: a binary regressor indicating whether the video includes primate content or

not, a binary regressor indicating that a video is played forward or backward, the five repetitions (or two repetitions for humans) of the video-trials, physical location of the selected probe (left or right), time elapsed within session, chosen frame location, temporal similarity, perceptual similarity (SURF), temporal distance, and the response of the subjects (correct/incorrect).

## Model comparison

To obtain the best fit among these models, we used Bayesian Information Criterion (BIC) as a criterion for model selection among these four models. The formula for BIC is: BIC = $-2(logL)$ + num-Param* log(numObs), where L is the maximum likelihood for the model, and numParam and numObs represent the number of free parameters and the number of samples, respectively. We computed $\Delta$BIC as the strength of the evidence, which indicates the extent to which the selected model is superior to other models. Different ranges of $\Delta$BIC show different level of evidence: a value of $\Delta$BIC larger than two shows positive evidence, and a value of $\Delta$BIC larger than six indicates strong evidence (*Kass and Raftery, 1995*).

## Temporal similarity

For each trial, we calculated temporal similarity (TS) as an index of the discriminability of probe frames. Temporal similarity between two probe frames extracted from the video is calculated by the ratio of the two frames' temporal separation between their occurrence in the video and the time of testing. The temporal similarity of any two memory traces can be calculated as: TS = delay2/delay1, where delay2 < delay1 (*Brown et al., 2007*).

## Perceptual similarity

We made use of three main parameters to measure the perceptual dissimilarity between TOJ frames for the GLMs. First, RGB-histogram is computed as the Sum-of Square-Difference (SSD) error between image pairs for the three color channels (RGB). For each color channel, the intensity values range from 0 to 255 (i.e., 256 bins), and we computed the total number of pixels at each intensity value and then the SSD for all 256 bins for each image pair. The smaller the value of the SSD, the more similar the two images (image pair) were. Second, for HOG similarity, we constructed a histogram of directions of gradient over fixed-sized grids across the entire image. A vector is generated from each grid cell and correlated with HOG features from another image. Third, Speeded Up Robust Features (SURF) (*Bay et al., 2008*) uses Box Filter using integral images (*Viola and Jones, 2004*) to approximate Laplacian-of-Gaussian (LoG). Wavelet responses in both horizontal and vertical directions are used to assign orientation in SURF. SURF consists of fixing a reproducible orientation that is based on information from a circular region around the interest point. A descriptor vector is generated around the interest point using the integral image, which matches with descriptor vectors extracted from a compared image. SURF uses various scales and different orientation to identify unique features or key-points in an image. If the same feature exits in another image that is smaller or larger in size or even at a different orientation, SURF identifies that feature (or key-point) as corresponding or similar in both images (*Bay et al., 2008*; see *Figure 6—figure supplement 1C* for illustration). The Euclidean distance is used to measure the similarity between two descriptor vectors from images.

## Additional information

### Funding

| Funder | Grant reference number | Author |
|---|---|---|
| 973 Program | 2013CB329501 | Yong-di Zhou |
| Ministry of Education of the People's Republic of China | Humanities and Social Sciences 16YJC190006 | Sze Chai Kwok |

The funders had no role in study design, data collection and interpretation, or the decision to submit the work for publication.

## Author contributions
Shuzhen Zuo, Conceptualization, Formal analysis, Investigation, Methodology, Writing - original draft, Writing - review and editing, Data interpretation; Lei Wang, Conceptualization, Investigation, Methodology, Data interpretation; Jung Han Shin, Software, Methodology, Data interpretation; Yudian Cai, Investigation, Methodology; Boqiang Zhang, Conceptualization, Resources, Data curation, Investigation; Sang Wan Lee, Methodology, Data interpretation; Kofi Appiah, Software, Methodology; Yong-di Zhou, Funding acquisition, Data interpretation; Sze Chai Kwok, Conceptualization, Supervision, Funding acquisition, Investigation, Methodology, Writing - original draft, Project administration, Writing - review and editing, Data interpretation

## Author ORCIDs
Shuzhen Zuo (ID) https://orcid.org/0000-0002-8917-8352
Lei Wang (ID) https://orcid.org/0000-0002-6224-6474
Jung Han Shin (ID) https://orcid.org/0000-0002-8237-2144
Sang Wan Lee (ID) http://orcid.org/0000-0001-6266-9613
Sze Chai Kwok (ID) https://orcid.org/0000-0002-7439-1193

## Ethics
Human subjects: The experimental protocol was approved by the the University Committee on Human Research Protection (permission code: HR 023-2017) at East China Normal University. The participants provided informed consent.
Animal experimentation: The experimental protocol was approved by the Institutional Animal Care and Use Committee (permission code: M020150902 & M020150902-2018) at East China Normal University. All experimental protocols and animal welfare adhered with the "NIH Guidelines for the Care and Use of Laboratory Animals".

## Decision letter and Author response
Decision letter https://doi.org/10.7554/eLife.54519.sa1
Author response https://doi.org/10.7554/eLife.54519.sa2

---

# Additional files

## Supplementary files
• Source data 1. Source data for all figures and tables.

• Supplementary file 1. Data description tables used to illustrate all of the key variables contained in the 'Source Data 1.xlsx'.

• Supplementary file 2. For monkeys: one sample $t$-test results for the slopes of reciprocal latency as a function of temporal similarity, having entered a range of nuisance variables as regressor-of-no-interest. The three panels correspond to analyses performed using all trials (top), only correct trials (middle), and only incorrect trials (bottom). The same slope patterns were observed irrespective of correctness, as is consistent with the analyses of slopes of reciprocal latency as a function of chosen frame location for each monkey. Related to *Figure 2—figure supplement 1A*.

• Supplementary file 3. Hierarchical multiple regression results for individual monkeys (left panel) and for human participants (right panel) showing reaction time as a function of chosen frame location (CFL).

• Supplementary file 4. For human participants: one sample $t$-test results for the slopes of reciprocal latency as a function of temporal similarity (upper panel) and the slopes of reciprocal latency as a function of chosen frame location (bottom panel) against zero, after having entered a range of nuisance variables as regressor-of-no-interest. Related to *Figure 2—figure supplement 1B*.

• Supplementary file 5. LATER model fitting results of six monkeys and seven human participants. For ease of comparison, we computed the respective ΔBIC to index the strength of evidence for each model. Note that the model with the lowest BIC is the winning model. In all 6 monkeys, the shift model is superior to the other three models, whereas this effect is not consistent in the humans. Related to *Figure 5*.

• Transparent reporting form

### Data availability

All data is available at Dryad (https://doi.org/10.5061/dryad.3r2280gcc).

The following dataset was generated:

| Author(s) | Year | Dataset title | Dataset URL | Database and Identifier |
|---|---|---|---|---|
| Zuo S, Wang L, Shin JH, Cai Y, Zhang B, Lee SW, Appiah K, Zhou Y, Kwok SC | 2020 | Behavioral evidence for memory replay of video episodes in macaque monkeys | http://doi.org/10.5061/dryad.3r2280gcc | Dryad Digital Repository, 10.5061/dryad.3r2280gcc |

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
