## [Decision Letter]

**Acceptance summary:**

Memory replay enables humans recall the past in a flexible and temporally-compressed manner. The extent to which this process is shared with nonhuman primates is unknown. Kwok and colleagues demonstrate that, like humans, macaque monkeys temporally compress past experiences with a non-linear forward-replay mechanism. However, replay in macaques was strictly forward; unlike humans, macaque monkeys lacked a global compression mechanism that enabled the flexibility to skip irrelevant information. This work helps to map the evolution the episodic memory.

**Decision letter after peer review:**

Thank you for submitting your article "Behavioral evidence for memory replay of video episodes in macaque monkeys" for consideration by *eLife*. Your article has been reviewed by three peer reviewers, and the evaluation has been overseen by a Reviewing Editor and Joshua Gold as the Senior Editor. The following individual involved in review of your submission has agreed to reveal their identity: Bryan Strange (Reviewer #3).

The reviewers have discussed the reviews with one another and the Reviewing Editor has drafted this decision to help you prepare a revised submission.

All reviewers agreed that this is an interesting and timely paper, and they universally congratulated you on the Herculean effort involved in conducting this research. The reviewers did raise several issues that must be addressed prior to publication. Their reviews are included below.

The manuscript presents a number of very complex analyses, which were at times difficult to digest. In the revision, we encourage the authors to improve the overall clarity of the manuscript by providing more (and clearer) details on the methods. Each reviewer has highlighted specific requests for more information and additional analyses to clarify results. Clearer linkage between the Materials and methods and corresponding Results would also help to improve the manuscript's clarity.

The reviewers raised questions about the behavioural performance measures (e.g., the need to more carefully consider the accuracy data, the inclusion of both correct and incorrect trials, and how stimulus repetition may affect the results). In several instances, the reviewers suggested additional interpretations of the data.

We hope that these comments are helpful as you prepare your revision, and look forward to receiving the revised manuscript.

*Reviewer #1:*In this manuscript, Zuo and colleagues present novel behavioural data from macaque monkeys with the aim of investigating the presence of replay during temporal order judgments for previously seen video clips. The question and paradigm are interesting and timely. The comparisons between monkey and human strategies and performance are especially intriguing. However, I have some questions about the behavioural performance and measures. The statistical models should also be reported more clearly and the link to predictions should be more explicit.

1) Additional details about the testing paradigm and overall behavioural results are needed. Were the monkeys trained ahead of time (and if so, how long did the training take)? With such a protracted period, it would be interesting to know whether the monkeys' performance improved or changed over the course of the experiment. As their performance is not at ceiling level, it seems possible that they improve overall, or that they start relying on different aspects to make their judgments. Day of testing could therefore also be included as a predictor in the models.

Regarding the monkeys' accuracy: in the subsection “Task performance”, the authors report that the monkeys' overall accuracy was ~68%. Was this the overall accuracy calculated across all 50 testing days? In contrast, the monkeys' accuracy plotted in Figure 2C is barely above chance level – while it's clear that the pattern displayed in the figure will be noisy as the data are split according to target frame location, the lines hover close to chance level and don't seem to average out to 68%. I may be missing something, but unless the data in Figure 2C represent only a subset, for example, this needs to be explained in more detail. Related to the point above, it would generally be helpful if the authors provided a measure of the monkeys' accuracy over the course of testing.

This is also important as the human participants' accuracy was at ceiling after a single testing session (based on the aforementioned subsection) – did the trend of the monkeys' performance remain the same across the 50 testing days (i.e., no global compression), or did this only develop after extensive experience on the task?

2) On a related note, it seems confusing to include both correct and incorrect trials in most of the analyses. In the GLM analyses (subsection “Human-like forward replay in macaques”, Supplementary file 2), the patterns were indeed quite similar for correct and incorrect trials, but it still seems unusual to include trials where the monkeys may not have replayed the content correctly as they reached an incorrect decision. Further, the different amounts of data entered into different models (correct trials only vs. all trials) make it more difficult to compare them. I would recommend that only correct trials are included in all analyses, or additional control analyses are needed to ensure that all findings remain the same when only correct trials are used.

3) The comparison between monkey and human data in Figure 3 is especially interesting. It is intriguing to see that human participants display global compression, but monkeys do not, and this interpretation seems to be well-supported by the data. However, the authors state that monkeys 'reach their memory decision threshold more quickly when probe frames are extracted from the two different contexts' and that these findings 'parallel closely established findings in the humans'. This pattern is in fact the opposite to that observed in humans; when two stimuli are separated by a boundary, humans are slower to reach a decision (e.g. Ezzyat and Davachi, 2014; Heusser et al., 2018, to name just two recent papers). This interpretation should be revised as the data from the present manuscript in fact suggest that boundaries affect human and monkey decision making processes in opposite directions.

4) The manuscript contains a relatively large number of (at times complex) analyses. As the more complex models are such an important aspect of the paper, they should be explained in more detail and the link between Materials and methods and Results should be clear. I found it somewhat difficult to align the description of the analytic approach in the Materials and methods section with the reporting of the results. In general, more detail in the description of the methods is needed. This especially applies to the model and variable descriptions. A few key points below:

– In the subsection “Task and experimental procedure”, the authors state that the experiment comprised four factors: boundary, play order, temporal distance, and exposure. First, I found it unusual that the authors reported temporal distance as a factor with 25 levels, since modeling it as a categorical variable assumes that all 25 levels are independent of one another. If this was not the case, it should be clarified in text, but if this variable was indeed modeled as a categorical factor, this should be changed to continuous or ordinal.

Second, in the subsection “Generalized linear models (GLM)”, the authors then report a set of variables included in the GLM. If I'm not mistaken, these are only regressed out of the very last GLM (subsection “Confirmatory GLMs for the putative patterns”). While the evidence from this analysis is clear and the significance of the key factors of interest does not change, I think including these regressors in each of the analyses and treating them as nuisance regressors would be more convincing/parsimonious.

– The description of the LATER model fitting was somewhat confusing. For example, it wasn't immediately clear that 'both conditions' (subsection “LATER (linear approach to threshold with ergodic rate) modelling”) refers to within vs. across clips. I also found it somewhat confusing that one of the models was called 'unconstrained' in the Materials and methods, but 'two fits' in Supplementary file 5 (if that is indeed the same model). Finally, in the Materials and methods, the authors refer to Figure 4A, which I believe should be Figure 6A. I am not a modeler, but it would nonetheless be important for the model descriptions to be linked back to the experimental predictions to make the connection between model parameters and behaviour clearer.

– How exactly was reciprocal latency calculated? I was not familiar with the term beforehand so I looked it up in the literature, but I would suggest that all key variables are defined in an accessible manner.

5) Related to the above: in the Introduction, the authors say that a 'non-linear' pattern would be predicted, which led me to expect a comparison of linear and non-linear model fits. Similarly, in the Materials and methods, the authors state that they used BIC to 'obtain the best fit among these models' – as this section immediately follows the section on GLMs, I assumed the models would be compared. However, the only BIC value reported is that comparing the 'shift' and 'swivel' models. I believe BIC can be used to compare any model types (i.e. linear and non-linear), but if this is not the case, the approach should be clarified. Unless I'm missing something, it appears that the reciprocal latency was modelled in a linear model (subsection “Human-like forward replay in macaques”), but the 'non-linearity' of the fit was only assessed visually. As the linear analyses all had significant outcomes, it seems important to provide a benchmark of 'goodness of fit' for different models. Reporting the significance of different trends (e.g. linear, quadratic, cubic) could be helpful.

*Reviewer #2:*In this study, Zuo et al. examined existence of memory replay during the retrieval of video clip material encoded continuously during periods of ~10 seconds. The study combined several computational modeling approaches and complex analytical procedures on reaction time data to test whether monkeys showed forward replay of the encoded material and whether the replayed memory content showed a structured pattern modulated by context changes during encoding, as shown previously in humans. Their results, though similar with humans in many aspects, also revealed striking differences, being the Inability to skip irrelevant Information In their replay a major one.

I found the study very well written and the topic of research of interest among many in the neuroscientific community. The analytical approach is sophisticated and the implementation sounding. I do have however some concerns that would require further attention though, which are listed below:

1) Many of the findings described in human studies related to sequence of images or video clips that are presented only once. Here, animals are shown repeatedly with the video clips. To what extent the repetition is affecting the underlying neural mechanisms and consequently the actual findings? I am aware the study included somehow this in their GLM model (Supplementary Figure 3) but in my view, this should require some more work. The question that one would like to see answered here would be: To what extent differences between humans and macaques are susceptible to be affected by the large amount of repeated material used in monkeys? I am aware many of the analysis included in the study require large number of trials for each individual but maybe authors can explore this issue across participants?

2) Several details concerning the experimental design tested in humans are lacking. Which are the differences between the two species when it comes to the experimental design? This information should be clear in the manuscript so that differences and similarities between species can be fully evaluated.

3) I was expecting that most of the analysis were implemented in monkeys' and in humans' data. Is there any particular reason to skip some of them in humans RTs (for example: effects of context change (within vs. across) and GLM confirmatory analysis)?

4) Correlation results between models should be directly compared and show they differed significantly to be able to attribute a winner one (i.e., Figure 3).

5) I found the results showing that many of the effects were equally robust for correct and incorrect trials a bit confusing. In my understanding, the behavioural manifestation of how memory content is organized and replayed should be specifically evident for when retrieval access has been successful, as otherwise it may be difficult to discard the possibility that the observations are driven by a more general task oriented operation. Can the authors please justify why it would be relevant that many of their central findings were valid for correct and incorrect trials? And if so, wouldn't it be also relevant to show the same results in humans' data?

*Reviewer #3:*By training 6 macaques on a cinematic video-clip task, Kwok and colleagues have leveraged reaction time (RT) data to make inferences on the ability of non-human primates to make temporal order judgment (TOJ). RT analyses, using LATER and drift-diffusion modelling, enabled to authors to suggest potential mechanistic underpinnings to the behavioural effects they observe. Irrespective of performance, RTs were faster if the still pertained to earlier segments of the video clip. The effect was non-linear, as the correlation was significant for log-transformed RTs. Furthermore, the relationship of still presentation latency to RT was around 10:1, which the authors interpret as time compression in replay. Humans, on the other hand do not show this relationship.

This is an interesting paper, and the cross-species differences are very clearly depicted and highly interesting. I commend the authors on what is a tremendous effort in terms of stimulus preparation and animal training. I have some comments that would need to be addressed before recommending publication.

1) The authors have concentrated mainly on RTs, but when accuracy is considered (plotted in Figure 2C) it is clear that for many target frame locations, the macaques are performing at chance (horizontal blue line). In at least 3 of the macaques, accuracy seems worse for earliest stills from clip 1. Could there be a speed-accuracy trade-off underlying faster RTs for these early stills? I appreciate that, overall, performance was above chance (it is somewhat atypical to report this at the beginning of the Materials and methods section), but the possible confound of monkeys making fast responses with little memory content to these early still probes needs addressing.

2) Furthermore, there is clearly an improvement in accuracy for stills that are at the beginning of clip 2. The authors mention this as "a blip" but provide no statistics. This is an interesting boundary effect that could be reported better and integrated with point 3.

3) Representational similarity analyses were used to demonstrate that "global compression" of individual video clips is not evident in macaques, who appear to show increasing RTs to stills drawn from over the course of the 2 clips (i.e. the strictly forward model). There is a non-significant trend towards global compression effects in humans. It is clear, though, that macaques and humans respond differently. What makes things a bit confusing is that LATER modeling indicated that macaques show an important boundary effect. Memory decision threshold is reached more quickly if probe stills come from different clips. I wonder whether this has something to do with perceptual similarity effects reported later for the GLM analyses, but I did not get a good feel for what this perceptual similarity parameter is measuring.

4) I am undecided as to whether Figure 5 and associated Results section really add to the findings. It is clear from the preceding figures that slope is markedly different in the two species.

5) In view of the indirect nature of the inference, certain statements such as "The monkeys apply a non-linear forward, time-compressed replay mechanism during the temporal-order judgement” (Abstract) need to be toned down.

[Editors' note: further revisions were suggested prior to acceptance, as described below.]

Thank you for resubmitting your article "Behavioral evidence for memory replay of video episodes in the macaque" for consideration by *eLife*. Your revised article has been reviewed by three peer reviewers, and the evaluation has been overseen by a Reviewing Editor and Joshua Gold as the Senior Editor The following individual involved in review of your submission has agreed to reveal their identity: Bryan Strange (Reviewer #3).

The reviewers have discussed the reviews with one another and the Reviewing Editor has drafted this decision to help you prepare a revised submission.

The reviewers were very satisfied with the revisions and we would like to congratulate you on a fine paper. The number of testing days for each animal is testimony to Herculean effort put into this work, and we believe that it will be a very important contribution to understanding cross-species differences in memory replay.

It was noted that with the additional analyses, the paper now shows more robust evidence for a forward scanning strategy in non-human primates. The additional analyses regarding training/testing session over time were also appreciated, in particular including response accuracy as a covariate and the trend analysis.

Only a few relatively minor comments remain:

1) The observation that the findings largely hold up when the data are split by correct and incorrect trials (Table 1) certainly supports the authors' decision to include all trials in their analysis, regardless of correctness. However, since the findings are so similar for correct and incorrect trials, the authors may wish to discuss why this pattern might be observed even on incorrect trials. Is the assumption that the monkeys are replaying the content correctly but then reaching an incorrect decision or that the information was incorrectly encoded? In other words, if the latency data reflects memory processes, an incorrect decision here would suggest that the initially encoded temporal order was incorrect. Either way, this seems like an interesting finding and Discussion point.

2) Regarding the analysis of linear, quadratic, and cubic trends: it is indeed encouraging to see that the non-linear (quadratic and cubic) trends are significant in the monkeys. However, interestingly, only the cubic trend seems to be significant in the human sample (linear and quadratic are not). Since one of the important contributions of this paper is a direct comparison between monkeys and humans, we think it would be helpful if the authors also addressed this difference in the manuscript. We also suggest that the manuscript text more explicitly stated what type of non-linear relationship was observed (i.e., in the subsection “Human-like forward replay in macaques” where these results are reported).

There were two requests for greater clarity:

1) The Introduction sets up the notion of linear vs. non-linear models for RTs and the authors state that they adjudicate between the two aspects of the replay models comparing between human and macaque data. While I appreciate that the non-linear human component refers to the global compression, the fact that monkeys appear to have performed TOJ using a forward search with non-linear compression, might confuse some readers. It was recommended that this be made explicit in the Introduction.

2) The legend of Figure 6—figure supplement 1 needs to include more explanation; please correct. Is the point of this to indicate that there is not a direct mapping of all features between inset and T-shirt images (i.e. the coloured lines don't always go to the same point of the corresponding image)?

---

## [Author Response]

Reviewer #1:[…]1) Additional details about the testing paradigm and overall behavioural results are needed. Were the monkeys trained ahead of time (and if so, how long did the training take)? With such a protracted period, it would be interesting to know whether the monkeys' performance improved or changed over the course of the experiment. As their performance is not at ceiling level, it seems possible that they improve overall, or that they start relying on different aspects to make their judgments. Day of testing could therefore also be included as a predictor in the models.

We now provided more details on these aspects. The monkeys had received training before the main test reported in the manuscript. There were five stages of learning on the TOJ task and the numbers of days are reported in (Table 2). With these extended periods of training, we think that the monkeys were unlikely to change over the course of the experiment in terms of their base TOJ ability (see also our reply below). In order to check this, we plotted out the proportion correct per day per individual: (A) Monkey data. (B) Human data. We observe no obvious increase in performance over the course of testing days for both monkeys and humans (Figure 7). This absence of performance changes is due to the extensive training the monkeys had received prior to this experiment (~ 21150 trials in total per monkey across 141 sessions on average, see Table 2). We therefore deem the main effects reported here as unlikely attributable to behavioral or strategic changes over the course of main experiment.

Regarding the monkeys' accuracy: in the subsection “Task performance”, the authors report that the monkeys' overall accuracy was ~68%. Was this the overall accuracy calculated across all 50 testing days? In contrast, the monkeys' accuracy plotted in Figure 2C is barely above chance level – while it's clear that the pattern displayed in the figure will be noisy as the data are split according to target frame location, the lines hover close to chance level and don't seem to average out to 68%. I may be missing something, but unless the data in Figure 2C represent only a subset, for example, this needs to be explained in more detail.

The results displaying ~68% are the aggregated data from all trials (both within- and across-) from all 50 testing days, whereas the data in Figure 2C contained only within-context trials. These are now clarified more explicitly in the revised manuscript (subsection “Human-like forward replay in macaques”). The analyses in this paper are broadly made of two parts: using only within-context trials and comparing within-context vs. across-context trials.

Related to the point above, it would generally be helpful if the authors provided a measure of the monkeys' accuracy over the course of testing. This is also important as the human participants' accuracy was at ceiling after a single testing session (based on the aforementioned subsection) – did the trend of the monkeys' performance remain the same across the 50 testing days (i.e., no global compression), or did this only develop after extensive experience on the task?

We plotted the accuracy data for each testing day for each monkey and human as Figure 7. We do not observe any obvious increase in performance over the course of the 50 testing days. Together with the extensive training sessions/trials prior to the experiment (over 20,000 trials per monkey). We deem the main effects reported in the manuscript as not attributable to changes over the course of main experiment.

Moreover, to further address the possibility that the two species might be differentially susceptible to the effect of having different numbers of trials in the experiment, we performed a control analysis making use of only the first 2 repetitions of data to equate the number of trials between the two species. With these equated subsets of data, we re-calculated the correlation between monkey RT RDM and two hypothetical model (i.e., Strict forward model vs. Global compression model). The results replicated the same significant correlation with the Strict forward model (r = 0.56, P = 0.049), but not with the Global compression model (r = -0.10, P = 0.643). These results replicated the findings as of when the full set of data was used, suggesting that the more numerous exposure to the same stimuli did not affect the main results. These results are now added in the revised manuscript (subsection “Discrepancy with humans: Compression of replay is local but not global”).

2) On a related note, it seems confusing to include both correct and incorrect trials in most of the analyses. In the GLM analyses (subsection “Human-like forward replay in macaques”, Supplementary file 2), the patterns were indeed quite similar for correct and incorrect trials, but it still seems unusual to include trials where the monkeys may not have replayed the content correctly as they reached an incorrect decision. Further, the different amounts of data entered into different models (correct trials only vs. all trials) make it more difficult to compare them. I would recommend that only correct trials are included in all analyses, or additional control analyses are needed to ensure that all findings remain the same when only correct trials are used.

In order to clarify this issue, we first made use of the GLMs, but now adding a new binary regressor (response: correct or incorrect) into the full GLMs. In this way, we checked how correctness might influencethe overall response latency per condition per species.The results indicate that response correctness influences the overall response latency considerably especially in the across-context condition in both species, but to a lesser extent for within-context condition in the monkeys (Figure 6). However, in order to directly address the main issue of “changes in response latency as a function of chosen frame location”, we improved the linear regression analyses by including other variables as nuisance regressors (including response correctness) (cf. reviewer 1 comment 4 below). We found a significantly positive relationship between RT and chosen frame location in all monkeys, all P < 0.001 (more significant than without the nuisance regressors). We have accordingly modified Table 1 (top panel) with these new results. Moreover, we have also conducted these analyses considering only correct trials (or only incorrect trials) separately. The new results are now also reported in Table 1 (middle and bottom panels respectively) (subsection “Human-like forward replay in macaques”). These results support that possibility that the retrieval RT patterns as function of chosen frame location are not affected by the memory outcome. The same has been done with response latency/temporal similarity slope analysis too (Supplementary file 2).

3) The comparison between monkey and human data in Figure 3 is especially interesting. It is intriguing to see that human participants display global compression, but monkeys do not, and this interpretation seems to be well-supported by the data. However, the authors state that monkeys 'reach their memory decision threshold more quickly when probe frames are extracted from the two different contexts' and that these findings 'parallel closely established findings in the humans'. This pattern is in fact the opposite to that observed in humans; when two stimuli are separated by a boundary, humans are slower to reach a decision (e.g. Ezzyat and Davachi, 2014; Heusser et al., 2018 to name just two recent papers). This interpretation should be revised as the data from the present manuscript in fact suggest that boundaries affect human and monkey decision making processes in opposite directions.

We have now added an analysis on the human data with the LATER and did not observe the same pattern as the monkeys (Figure 5C and Supplementary file 3). It is worth mentioning that in paradigms making use of discreet images for encoding, the TOJ performance thus far reported in the human literature is distinct from those making use of continuous streaming video material. Specifically, it has been reported that accuracy and response times are worse in TOJ for items are that separated by a boundary during encoding in the literature (Ezzyat and Davachi, 2014; Heusser et al., 2018), whereas we show that TOJ performance is better between frames taken from an across-context condition than from a within-context condition (Figure 1—figure supplement 1A). This is likely due to the ease of remembering ‘discrete segment of events’ when making use of the contextual or perceptual clues in the video frames in an across-context condition (see also Figure 6—figure supplement 1 that contextual information affects TOJ). We have added this discussion briefly in the revised manuscript (subsection “Human-like forward replay in macaques”).

4) The manuscript contains a relatively large number of (at times complex) analyses. As the more complex models are such an important aspect of the paper, they should be explained in more detail and the link between Materials and methods and Results should be clear. I found it somewhat difficult to align the description of the analytic approach in the Materials and methods section with the reporting of the results. In general, more detail in the description of the methods is needed. This especially applies to the model and variable descriptions. A few key points below:

Thank you for this suggestion. Apart from addressing the specific sub-points mentioned below, we have re-organized some content from the Materials and methods section and provided some more detailed clarification of the methods. For example, to clarify our methods in parameterizing perceptual similarity information between TOJ frames, we have provided an illustration of what the main perceptual similarity parameter indexes in Figure 6—figure supplement 1C.

– In the subsection “Task and experimental procedure”, the authors state that the experiment comprised four factors: boundary, play order, temporal distance, and exposure. First, I found it unusual that the authors reported temporal distance as a factor with 25 levels, since modeling it as a categorical variable assumes that all 25 levels are independent of one another. If this was not the case, it should be clarified in text, but if this variable was indeed modeled as a categorical factor, this should be changed to continuous or ordinal.

Temporal distance is an ordinal variable. We have clarified this in the revised manuscript (subsection “Task and experimental procedure”).

Second, in the subsection “Generalized linear models (GLM)”, the authors then report a set of variables included in the GLM. If I'm not mistaken, these are only regressed out of the very last GLM (subsection “Confirmatory GLMs for the putative patterns”). While the evidence from this analysis is clear and the significance of the key factors of interest does not change, I think including these regressors in each of the analyses and treating them as nuisance regressors would be more convincing/parsimonious.

We now ran the linear regression analysis of response latency as a function of chosen frame location/temporal similarity having included other variables as nuisance regressors for each monkey separately. We now found a significantly positive relationship between RT and chosen frame location in all monkeys, all P < 0.001 (more significant than without these nuisance variables). We have modified Table 1 (top panel) with these new results. Moreover, related to comment 2 above, we have also conducted these analyses considering only correct trials (and only incorrect trials) separately. The results are reported in Table 1 (middle and bottom panels respectively) (subsection “Human-like forward replay in macaques”). We also ran the same sets of analyses on human participant data for comparison between the two species. The results are reported in Supplementary file 4.

– The description of the LATER model fitting was somewhat confusing. For example, it wasn't immediately clear that 'both conditions' (subsection “LATER (linear approach to threshold with ergodic rate) modelling”) refers to within vs. across clips. I also found it somewhat confusing that one of the models was called 'unconstrained' in the Materials and methods, but 'two fits' in Supplementary file 5 (if that is indeed the same model). Finally, in the Materials and methods, the authors refer to Figure 4A, which I believe should be Figure 6A. I am not a modeler, but it would nonetheless be important for the model descriptions to be linked back to the experimental predictions to make the connection between model parameters and behaviour clearer.

We apologize for the confusion and the mistaken reference. To clarify, the ‘unconstrained’ model is the same as ‘two fits’ model, and we have now only used the term ‘two fits’ model throughout the main text and the table. We have also corrected the reference to correct figure in the revised manuscript.

– How exactly was reciprocal latency calculated? I was not familiar with the term beforehand so I looked it up in the literature, but I would suggest that all key variables are defined in an accessible manner.

Reciprocal latency is the multiplicative inverse of raw reaction time (i.e., 1/RT). We added its definition in the Materials and methods (subsection “LATER (linear approach to threshold with ergodic rate) modelling”).

5) Related to the above: in the Introduction, the authors say that a 'non-linear' pattern would be predicted, which led me to expect a comparison of linear and non-linear model fits. Similarly, in the Materials and methods, the authors state that they used BIC to 'obtain the best fit among these models' – as this section immediately follows the section on GLMs, I assumed the models would be compared. However, the only BIC value reported is that comparing the 'shift' and 'swivel' models. I believe BIC can be used to compare any model types (i.e. linear and non-linear), but if this is not the case, the approach should be clarified. Unless I'm missing something, it appears that the reciprocal latency was modelled in a linear model (subsection “Human-like forward replay in macaques”), but the 'non-linearity' of the fit was only assessed visually. As the linear analyses all had significant outcomes, it seems important to provide a benchmark of 'goodness of fit' for different models. Reporting the significance of different trends (e.g. linear, quadratic, cubic) could be helpful.

We formally tested for any ‘non-linearity’ of the fit. We ran curvilinear regression analyses using reaction time as the dependent variable as chosen frame location to estimate the significance of different trends. We started testing for the linear relationship and progressively moved to higher-order relationships (quadratic, cubic). At each step, we assessed the significance of the new predictor, having accounted for all the variance fitted by the lower-level predictors. We reported the significance of different trends (linear, quadratic, cubic) for the two species (Supplementary file 5). This non-linear relationship is important because it rules out alternative explanations that the main effects are simply resultant from positional effects. Rather, the non-linear change in slope indicates there are other factors in play that the monkeys might group (or parse) the content according to the relational storyline structure (e.g., the 2-clip video design), and do not merely recall the frames/items as of their ordinal positions in a fixed linear manner.

Reviewer #2:[…]1) Many of the findings described in human studies related to sequence of images or video clips that are presented only once. Here, animals are shown repeatedly with the video clips. To what extent the repetition is affecting the underlying neural mechanisms and consequently the actual findings? I am aware the study included somehow this in their GLM model (Supplementary Figure 3) but in my view, this should require some more work. The question that one would like to see answered here would be: To what extent differences between humans and macaques are susceptible to be affected by the large amount of repeated material used in monkeys? I am aware many of the analysis included in the study require large number of trials for each individual but maybe authors can explore this issue across participants?

In a control analysis, we now made use of only the first 2 repetitions of data to equate the number of trials between the two species (note that the same experimental design was adopted by the human participants, see our reply to comment 2 below). Now with these equated subsets of data, we re-calculated the correlation between monkey RT RDM and two hypothetical model (i.e., Strict forward model vs. Global compression model). The results replicated the significant correlation with Strict forward model (r=0.56, P=0.049), but not with the Global compression model (r=-0.10, P=0.643). With this smaller set of data, we also directly compared the correlations between the two models and found significant differences between them (P<0.05, FDR-corrected, cf. reply to comment 4 below). These results replicated the findings as of when the full set of data was used, suggesting that having the more numerous exposure to the same stimuli did not affect the main results. These results are now added in the revised manuscript (subsection “Discrepancy with humans: Compression of replay is local but not global”).

It may be worth mentioning that despite having administered a large amount of training to the animals prior to this study, the monkeys’ performance have stabilized over the course of the experiment (see our reply to reviewer 1, point 1 and revised Materials and methods, subsection “Training history and task performance”). We have also taken care to ensure that all the videos used in the training stages were sampled differently from those employed in the main test, and that each of the monkeys was assigned with a pseudo-randomized remix of a unique 1000 video-trials set from the 2000-video library, thus eliminating video idiosyncrasy to be associated with any experimental conditions. With the large pool of unseen videos we have generated, only five repeated presentations were needed to accrue the relatively large numbers of trials (5000 per animal), which we believe is not too deviant from accepted standards in psychological studies in human.

2) Several details concerning the experimental design tested in humans are lacking. Which are the differences between the two species when it comes to the experimental design? This information should be clear in the manuscript so that differences and similarities between species can be fully evaluated.

The experimental design is essentially the same as the monkeys. The human participants re-used the 6 unique video-trials/list sets and TOJ frames (one unique set per monkey) correspondingly (the extra subject 7 re-used set 1). The only critical difference was that human participants performed only two exposures (R1 – R2) rather than five. Identically, each session contained 100 trials and with 20 daily sessions of testing, we have acquired 2000 trials per participant for analysis. Human participants used the same model of computers and touch-sensitive screens as monkeys. The human subjects sat ~30cm in front of another identical 17-inch infrared touch-sensitive screen, with stimuli presented with the same computer used in the monkeys’ experiment. These details are now further clarified in the revised manuscript (subsections “Apparatus and testing cubicle”, “Source of video materials and preparation” and “Task and experimental procedure”).

3) I was expecting that most of the analysis were implemented in monkeys' and in humans' data. Is there any particular reason to skip some of them in humans RTs (for example: effects of context change (within vs. across) and GLM confirmatory analysis)?

We have now completed the full comparison between the two species’ data with a series of new analyses with human data. These include the following main analyses: (i) LATER modelling for within- vs. across-conditions (Figure 5 and Supplementary file 3) and (ii) GLM confirmatory analyses (Figure 6B and Figure 6—figure supplement 1B). In addition, we also completed the comparison on (iii) the relationship between temporal similarity and reciprocal latency for within-context trials in humans (Figure 2—figure supplement 1B), (iv) proportion corrects and RT analyses for the humans (Figure 1—figure supplement 1A and Figure 7). These altogether help elucidate the differences and similarities between the species more explicitly.

4) Correlation results between models should be directly compared and show they differed significantly to be able to attribute a winner one (i.e., Figure 3).

We used two-sided Wilcoxon signed-rank tests to test the null hypothesis that the correlation between data RDM and the two hypothetical models are equal. Pairwise comparisons between the two models are statistically significant for monkeys (P<0.01) and humans (P<0.05), confirming the significant differences between the models in both cases. These are now modified in the revised manuscript and in Figure 3 (subsections “Discrepancy with humans: Compression of replay is local but not global”, and “Representational similarity analysis (RSA)”).

5) I found the results showing that many of the effects were equally robust for correct and incorrect trials a bit confusing. In my understanding, the behavioural manifestation of how memory content is organized and replayed should be specifically evident for when retrieval access has been successful, as otherwise it may be difficult to discard the possibility that the observations are driven by a more general task oriented operation. Can the authors please justify why it would be relevant that many of their central findings were valid for correct and incorrect trials? And if so, wouldn't it be also relevant to show the same results in humans' data?

Making use of the GLMs, we now added a new binary regressor (response: correct or incorrect) into the full GLMs. In this way, we checked how correctness might influencethe overall response latency per condition per species.The results indicate that response correctness influences response latency considerably especially in the across-context condition in both species, but to a lesser extent for within-context condition in the monkeys (Figure 6). However, to directly address the main issue of “changes in response latency as a function of chosen frame location”, we also ran the linear regression analyses but now including other variables as nuisance regressors (including response correctness). We found a significantly negative relationship between reciprocal latency and chosen frame location in all monkeys, all P < 0.001 (more significant than without those nuisance variables). We have modified Table 1 (top panel) with these new results. Moreover, we have also re-conducted these analyses considering only correct trials (or only incorrect trials) separately. The results are now reported in Table 1 (middle and bottom panels respectively) (subsection “Human-like forward replay in macaques”). These results support the possibility that the response latency patterns are not affected by the memory outcome.

Reviewer #3:[…]1) The authors have concentrated mainly on RTs, but when accuracy is considered (plotted in Figure 2C) it is clear that for many target frame locations, the macaques are performing at chance (horizontal blue line). In at least 3 of the macaques, accuracy seems worse for earliest stills from clip 1. Could there be a speed-accuracy trade-off underlying faster RTs for these early stills? I appreciate that, overall, performance was above chance (it is somewhat atypical to report this at the beginning of the Materials and methods section), but the possible confound of monkeys making fast responses with little memory content to these early still probes needs addressing.

In order to rule out that the putative RT effects are not driven by a trade-off in accuracy, we segmented each video into four segments (i.e., each clip was parsed into two halves based on target frame location separately) and calculated the inverse efficiency score [RT (ms) / percentage correct (%)] for each segment per individual. The monkeys showed a mild numeral increase trend in inverse efficiency score across the four segments but did not reach statistical significance (Figure 1—figure supplement 1B, left panel) (F (3, 20) = 0.10, P = 0.96), suggesting that the increase in RT towards the end of the video did not contribute to better accuracy. In contrast, we found that the humans show a lower inverse efficiency score for the video parts right after a boundary (F (3, 24) = 4.17, P = 0.016), with post hoc tests showing significant difference between bars 2 and 3 (Figure 1—figure supplement 1B, right panel). This difference between bars 2 and 3 suggests that the participants are generally faster in their decision (while having accuracy controlled for) right after a clip boundary. This pattern of boundary effect aligns with the little “blip” in proportion correct right after the beginning of Clip 2 in the humans (Figure 2B, right panel). This discussion is now added in the revised manuscript (subsection “Human-like forward replay in macaques”).

2) Furthermore, there is clearly an improvement in accuracy for stills that are at the beginning of clip 2. The authors mention this as "a blip" but provide no statistics. This is an interesting boundary effect that could be reported better and integrated with point 3.

Together with an additional analysis making use of inverse efficiency score (cf. reply to comment 1), it is clear that humans can make use of the across-clip boundary to facilitate their TOJ. Interestingly, in paradigms making use of discreet images for encoding, the TOJ performance thus far reported in the human literature is distinct from those making use of continuous streaming video material. Specifically, accuracy and response times are worse in TOJ for items are that separated by a boundary during encoding (Ezzyat and Davachi, 2014; Heusser et al., 2018), whereas TOJ performance is better between frames taken from an across-context condition than from a within-context condition (Figure 1—figure supplement 1A). This might be due to the ease of remembering ‘discrete segment of events’ when making use of the contextual or perceptual cues in the video frames in an across-context condition (see also Figure 6—figure supplement 1 that contextual information affects TOJ). We have added this discussion briefly in the revised manuscript (subsection “Human-like forward replay in macaques”).

3) Representational similarity analyses were used to demonstrate that "global compression" of individual video clips is not evident in macaques, who appear to show increasing RTs to stills drawn from over the course of the 2 clips (i.e. the strictly forward model). There is a non-significant trend towards global compression effects in humans. It is clear, though, that macaques and humans respond differently. What makes things a bit confusing is that LATER modeling indicated that macaques show an important boundary effect. Memory decision threshold is reached more quickly if probe stills come from different clips. I wonder whether this has something to do with perceptual similarity effects reported later for the GLM analyses, but I did not get a good feel for what this perceptual similarity parameter is measuring.

Thank you for raising this consideration. While both species make use of the perceptual dissimilarity to some extent (see Figure 6—figure supplement 1), their benefits from utilizing the perceptual information seemingly differ. Specifically, we have now added an analysis on the human data with the LATER and indeed did not observe the same pattern as the monkeys (Figure 5C and Supplementary file 3; cf. reviewer 1’s comment 3), suggesting that the two species might not treat the information given by the event boundary at TOJ in the same manner. Moreover, to clarify our methods in parameterizing perceptual similarity information between TOJ frames, we have provided a more intuitive illustration of what the main perceptual similarity parameter (SURF) indexes in Figure 6—figure supplement 1.

4) I am undecided as to whether Figure 5 and associated Results section really add to the findings. It is clear from the preceding figures that slope is markedly different in the two species.

Thank you for this suggestion. Indeed, reviewer 1 also concurs with this concern. Removal of Figure 5 is justified given that most of the information conveyed by Figure 5 is shown in Figures 3 and 4.

5) In view of the indirect nature of the inference, certain statements such as "The monkeys apply a non-linear forward, time-compressed replay mechanism during the temporal-order judgement” (Abstract) need to be toned down.

Yes, indeed. Thank you for this suggestion. We have made changes in the revised manuscript to emphasize that the data (or analyses) patterns are suggestive of a possibility of memory replay in the animals, or similar statements to that effect (e.g., Abstract, Introduction, Results and Discussion).

[Editors' note: further revisions were suggested prior to acceptance, as described below.]

1) The observation that the findings largely hold up when the data are split by correct and incorrect trials (Table 1) certainly supports the authors' decision to include all trials in their analysis, regardless of correctness. However, since the findings are so similar for correct and incorrect trials, the authors may wish to discuss why this pattern might be observed even on incorrect trials. Is the assumption that the monkeys are replaying the content correctly but then reaching an incorrect decision or that the information was incorrectly encoded? In other words, if the latency data reflects memory processes, an incorrect decision here would suggest that the initially encoded temporal order was incorrect. Either way, this seems like an interesting finding and Discussion point.

Thank you for pointing this out. We are also intrigued by this finding. We have previously showed in analogous TOJ paradigms in humans that TOJ task specific BOLD activation and behavioural RT patterns are independent of retrieval accuracy (Ezzyat and Davachi, 2014; Heusser et al., 2018). We interpreted these effects as process-based rather than content-based. At present, the latency results likely also point towards some “search” or replay memory processes, and any ultimate incorrect responses would thus likely be caused by memory noises injected during encoding and/or during delay maintenance. We have now discussed these issues more explicitly, as evidence for cross-species correspondence, in the revised text (Discussion).

2) Regarding the analysis of linear, quadratic, and cubic trends: it is indeed encouraging to see that the non-linear (quadratic and cubic) trends are significant in the monkeys. However, interestingly, only the cubic trend seems to be significant in the human sample (linear and quadratic are not). Since one of the important contributions of this paper is a direct comparison between monkeys and humans, we think it would be helpful if the authors also addressed this difference in the manuscript. We also suggest that the manuscript text more explicitly stated what type of non-linear relationship was observed (i.e., in the subsection “Human-like forward replay in macaques” where these results are reported).

This analysis also helps highlight differences between humans and macaque monkeys. As we found that the human data fits best, and only, with the cubic model, it suggests humans might have treated the video and TOJ differently from the macaques. This difference is reflected in the global compression capability in the human participants. We link this part to the reporting of its subsequent section in the revised manuscript (subsection “Human-like forward replay in macaques”).

To stipulate the non-linear relationship more precisely, we have added “quadratic and cubic” to qualify the non-linear pattern in monkeys.

There were two requests for greater clarity:1) The Introduction sets up the notion of linear vs. non-linear models for RTs and the authors state that they adjudicate between the two aspects of the replay models comparing between human and macaque data. While I appreciate that the non-linear human component refers to the global compression, the fact that monkeys appear to have performed TOJ using a forward search with non-linear compression, might confuse some readers. It was recommended that this be made explicit in the Introduction.

We clarify this by stating that while both species do recall the video content non-linearly, there is an aspect of discrepancy between the two species, in which the monkeys do not compress the cinematic events globally as effectively as in humans, whereas human participants possess an ability to skip irrelevant information of the video. This is now clarified in the revised manuscript (Introduction).

2) The legend of Figure 6—figure supplement 1 needs to include more explanation; please correct. Is the point of this to indicate that there is not a direct mapping of all features between inset and T-shirt images (i.e. the coloured lines don't always go to the same point of the corresponding image)?

We modified the legend of Figure 6—figure supplement 1C. Specifically, we clarified that the cartoon is to illustrate that how features from two images can be matched irrespective of their scale.We can see that features of the inset are matched to the T-shirt on how strongly they are related, but some feature points (mostly minority) may still incorrectly match other parts of the image.